# Developing inhibitory peptides against SARS-CoV-2 envelope protein

**Ramsey Bekdash**[1,2,3‡], **Kazushige Yoshida**[1,2‡], **Manoj S. Nair**[4], **Lauren Qiu**[1,2,5,6],
**Johnathan Ahdout**[6], **Hsiang-Yi Tsai**[6], **Kunihiro Uryu**[7], **Rajesh K. Soni**[8], **Yaoxing Huang**[4],
**David D. Ho**[4,9,10], **Masayuki Yazawa**[1,2,3,6]*

1 Department of Rehabilitation and Regenerative Medicine, Columbia University, New York, New York, United States of America, 2 Columbia Stem Cell Initiative, Columbia University, New York, New York, United States of America, 3 Department of Pharmacology, Columbia University, New York, New York, United States of America, 4 Aaron Diamond AIDS Research Center, Columbia University, New York, New York, United States of America, 5 Department of Biological Science, Columbia University, New York, New York, United States of America, 6 Department of Pharmacological Sciences, Icahn School of Medicine at Mount Sinai, New York, New York, United States of America, 7 EMSCOPIC, New York, New York, United States of America, 8 Proteomics and Macromolecular Crystallography Shared Resource, Columbia University, New York, New York, United States of America, 9 Department of Microbiology and Immunology, Columbia University, New York, New York, United States of America, 10 Division of Infectious Diseases, Department of Medicine, Columbia University, New York, New York, United States of America

‡ These authors share first authorship on this work.
* masayuki.yazawa@mssm.edu

**Data Availability Statement:** All relevant data are within the paper and its Supporting Information files. The data can be also accessible through Icahn School of Medicine at Mount Sinai Data Management and Research Compliance

## Abstract

Severe Acute Respiratory Syndrome Coronavirus 2 (SARS-CoV-2) has affected approximately 800 million people since the start of the Coronavirus Disease 2019 (COVID-19) pandemic. Because of the high rate of mutagenesis in SARS-CoV-2, it is difficult to develop a sustainable approach for prevention and treatment. The Envelope (E) protein is highly conserved among human coronaviruses. Previous studies reported that SARS-CoV-1 E deficiency reduced viral propagation, suggesting that E inhibition might be an effective therapeutic strategy for SARS-CoV-2. Here, we report inhibitory peptides against SARS-CoV-2 E protein named iPep-SARS2-E. Leveraging E-induced alterations in proton homeostasis and NFAT/AP-1 pathway in mammalian cells, we developed screening platforms to design and optimize the peptides that bind and inhibit E protein. Using Vero-E6 cells, human-induced pluripotent stem cell-derived branching lung organoid and mouse models with SARS-CoV-2, we found that iPep-SARS2-E significantly inhibits virus egress and reduces viral cytotoxicity and propagation in vitro and in vivo. Furthermore, the peptide can be customizable for E protein of other human coronaviruses such as Middle East Respiratory Syndrome Coronavirus (MERS-CoV). The results indicate that E protein can be a potential therapeutic target for human coronaviruses.

## Introduction

The Coronavirus Disease 2019 (COVID-19) pandemic caused by the Severe Acute Respiratory Syndrome Coronavirus 2 (SARS-CoV-2) [1–3] has affected approximately 800 million people

Committee. All materials and plasmid constructs used in this study have been maintained by Dr. Yazawa's laboratory and are available upon request. In the future, the constructs will be deposited and available to order through Addgene.

**Funding:** This work was supported by Columbia University Dean's Office Fund and Columbia University Translational Therapeutics (TRx) Pilot Award (to M.Y.). The funders had no role in study design, data collection and analysis, decision to publish, or preparation of the manuscript.

**Competing interests:** M.Y., R.B., K.Y., D.D.H., M.S.N., and Y.H. (inventors) filed a patent (Attorney Docket No.: 01001/00889-US0; status: Filed, 04/13/2022) related to this manuscript. This patent is for using synthetic peptides targeting SARS-CoV-2 envelope protein for treating COVID-19 and related human coronaviruses. The rest of the authors declare no competing interests.

**Abbreviations:** BSA, bovine serum albumin; COVID-19, Coronavirus Disease 2019; DMEM, Dulbecco's Modified Eagle Media; DMSO, Dimethylsulfoxide; ECL, enhanced chemiluminescence; ER, endoplasmic reticulum; ERGIC, ER-Golgi inter-compartment; FBS, fetal bovine serum; FDR, false discovery rate; HEK, human embryonic kidney; KLH, keyhole limpet haemocyanin; MERS-CoV, Middle East Respiratory Syndrome Coronavirus; NCE, normalized collision energy; NIH, National Institute of Health; PBS, phosphate-buffered saline; PCR, polymerase chain reaction; PS, penicillin/streptomycin; PVDF, polyvinylidene difluoride; SARS-CoV-2, Severe Acute Respiratory Syndrome Coronavirus 2; SPS, synchronous precursor selection; TBS-T, Tris-buffered saline with 0.1% Tween 20; TMT, tandem mass tag.

and counting in the world. More than 6 million people have passed away due to the viral infection. Because the mutagenesis rate in SARS-CoV-2 genes such as Spike is high [4–6], it is a challenge to develop sustainable approaches for prevention and treatment. While several new vaccines and drug candidates have become available, the number of COVID-19 infections and deaths are still increasing, and new variants are being reported [7–10]. Therefore, it would be ideal if we could target coronavirus genes that are highly conserved in SARS-CoV-1 and SARS-CoV-2 variants, as well as other human coronaviruses. Human coronaviruses such as SARS-CoV-2 and Middle East Respiratory Syndrome Coronavirus (MERS-CoV) express an Envelope (E) protein that forms an ion channel essential for viral function called a viroporin [11–17]. E protein is known to induce cellular toxicity via a number of different molecules and signaling pathways [12,16,18–23] and E protein is also thought to be involved in the Spike protein protection and maturation in host cells [20,24]. Compared to the other molecules, E protein is highly conserved among coronaviruses: SARS-CoV-2 E (2E) protein's 75 amino acid residues have high homology with SARS-CoV-1 E protein (approximately 96%), with identical amino (N)-terminus, transmembrane, and pore structures while their carboxyl (C)-terminus is slightly different [11,25–31] (Fig 1A). Previous studies reported that deficiency of SARS-CoV-1 E gene significantly reduced viral propagation [12], suggesting that 2E may also play essential roles in viral function and can be a potential therapeutic target for COVID-19 and future variants [10]. Therefore, in this study we seek to develop screening platforms and applied them to identify drug candidates against 2E. In addition, we examined whether our therapeutic approach targeting 2E can be applicable to the E protein of other human coronaviruses.

## Results

Previous studies have reported oligomeric structures and molecular interactions in SARS-CoV-1 E and 2E [25,26,32]. Following the results, we hypothesized that the highly conserved N-terminal region is crucial for oligomerization and that the N-terminal fragment might be able to disrupt 2E protein oligomerization and function because the N-termini might interact with each other in the protein oligomers. Because our previous study demonstrates that the overexpression of 2E affects proton homeostasis in intracellular organelles such as Golgi apparatus and lysosomes in mammalian cells [19], we examined the effect of the N-terminal fragment, which is named MY18 (18 amino acids, MYSFVSEETGTLIVNSVL), on 2E function using DND-189 pH fluorescent dye and MY18 plasmid transfection in mammalian cells. DND-189-based pH fluorescent imaging shows that MY18 co-overexpression in 2E-expressing mammalian cells significantly restores DND-189 fluorescence to the normal level (Fig 1B and 1C). These results encouraged further investigation in using MY18 as a peptide that inhibits 2E. Next, to apply MY18 as an exogenous synthetic peptide, we tested 3 cell-penetrating amino acid motifs: an arginine repeat, TAT and Penetratin [33] (Fig 1D). The DND-189-based pH imaging demonstrates that the TAT version of MY18 is the most promising cell-penetrating peptide among the cell-penetrating peptide candidates (Fig 1E).

While DND-189-based pH fluorescent imaging is useful as a drug screening platform of live mammalian cells, the dynamic range of the dye is somewhat limited, the standard deviation of fluorescent readout is relatively large, and its throughput is not as high as that of an assay for screening. These limitations may give rise to difficulties in further optimizing the MY18 peptide using molecular biological approach with mutagenesis. To develop a higher-throughput screening platform, we overexpressed 2E in mammalian cells and explored other reliable and quantitative readouts. Interestingly, global proteomics results showed increases of various key signaling molecules such as JUN/AP-1 (Fig 2A). A follow-up experiment using

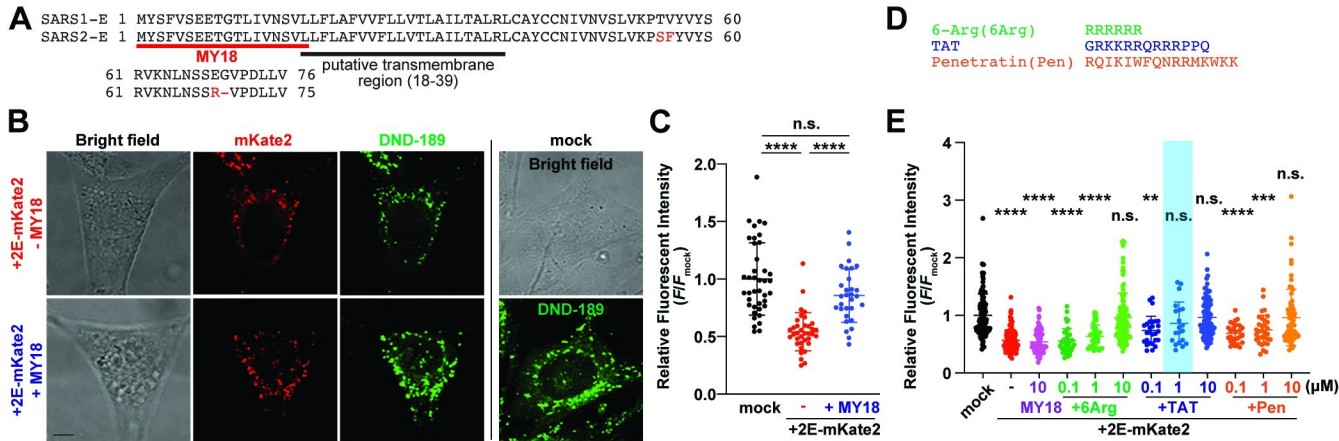

**Fig 1. Testing the amino-terminal fragment of SARS-CoV-2 Envelope named MY18 on lysosomal pH imaging in mammalian cells.** (**A**) Alignment of Envelope of SARS-CoV (SARS1-E) and SARS-CoV-2 (SARS2-E), showing the difference in amino acids (red), the targeted amino-terminal region named MY18 (red underlined), and putative transmembrane region (black underlined, 18–39). (**B**) Representative confocal fluorescent and bright field images of NIH 3T3 cells loaded with DND-189, a lysosomal pH green fluorescent dye, and transfected with MY18 peptide construct, empty vector (mock) and SARS2-E fused with mKate2 red fluorescent protein (2E-mKate2). Scale bar, 5 μm. (**C**) Relative fluorescent intensity of DND-189 dye in NIH 3T3 cells transfected using mock ($n = 40$) or 2E-mKate2 plasmid without (-, $n = 36$) and with MY18 plasmid (+MY18, $n = 30$). One-way ANOVA with Tukey's multiple comparisons test (**** $P < 0.0001$; n.s. not significant). (**D**) The sequences of cell-penetrating peptide candidates, 6-Arg (6Arg), TAT and Penetratin (Pen), for MY18 peptide uptake in mammalian cells. (**E**) Relative fluorescent intensity of DND-189 dye in NIH 3T3 cells incubated with MY18 (10 μM, $n = 93$), 6Arg-MY18 (0.1 μM, $n = 40$; 1, $n = 41$; 10, $n = 106$), TAT-MY18 (0.1 μM, $n = 27$; 1, $n = 20$; 10, $n = 92$), and Pen-MY18 peptides (0.1 μM, $n = 27$; 1, $n = 32$; 10, $n = 68$) with 2E-mKate2 plasmid transfection. Mock ($n = 116$) and non-treated 2E-mKate2 (-, $n = 139$) were also tested as their controls. In 1 μM condition, TAT-MY18 peptide is not different from mock (blue highlighted) while 6Arg and Pen are significantly lower than mock. One-way ANOVA with Dunnett's multiple comparisons test (**** $P < 0.0001$; *** $P < 0.001$; ** $P < 0.01$; n.s. not significant, compared to mock). The data underlying this figure can be found in S1 Data. All the graphs in the figure are mean ± SD. NIH, National Institute of Health; SARS-CoV-2, Severe Acute Respiratory Syndrome Coronavirus 2.

quantitative RT-PCR (qPCR) confirmed that the expression of most of the genes including *JUN* transcript are significantly increased in 2E-expressing cells compared to mock (Figs 2B and S1). The transcript expression of a *JUN*-related molecule, *NFATC4*/NFAT3, is also up-regulated significantly (Fig 2C), though the increase of *NFATC4*/NFAT3 protein (approximately 120%) did not reach to significance in statistical analysis of the global proteomics results with the standard false discovery rate (FDR, 0.05, Fig 2A). Following these results, we decided to apply the NFAT response element of the human *IL-2* gene where NFAT and JUN/AP-1 interact [34]. To obtain precise readouts, we used a dual luciferase reporter system containing NFAT *Firefly* luciferase reporter and pRL-TK-*Renilla* luciferase reporter as the transfection control using HSV *TK*, *herpes simplex virus thymidine kinase*, promoter [35] (Fig 2D). The luciferase assay result obtained using plasmid DNA co-transfection shows that 2E overexpression significantly increases the *Firefly* luciferase activity in mammalian cells and that MY18 co-expression significantly suppresses the effect of 2E on the NFAT/AP-1 pathway, though it does not fully restore it to mock levels (Fig 2E). These results suggest that MY18 is not sufficient to prevent 2E from altering the NFAT/AP-1 pathway completely, though we observed that MY18 restores proton homeostasis in DND-189-based pH fluorescent imaging (Fig 1). Looking to optimize MY18, we next examined whether deletion or extension of MY18 might improve the effect on interrupting 2E function using the luciferase reporter. We found that none of the constructs significantly improved the effect, as the majority reduced efficacy (Fig 2F). Next, we tested a variety of MY18 mutant constructs using mutagenesis, co-transfection and luciferase assay, and found that the substitution of glutamate to aspartate at seventh and eighth residues (EE7-8DD or 2ED) significantly improved MY18 (Fig 2G). We combined the mutant 2ED and TAT cell-penetrating motif (Fig 1D and 1E) and called TAT-MY18-2ED "iPep-SARS2-E" (inhibitory Peptide against SARS-CoV-2 Envelope). In addition, we

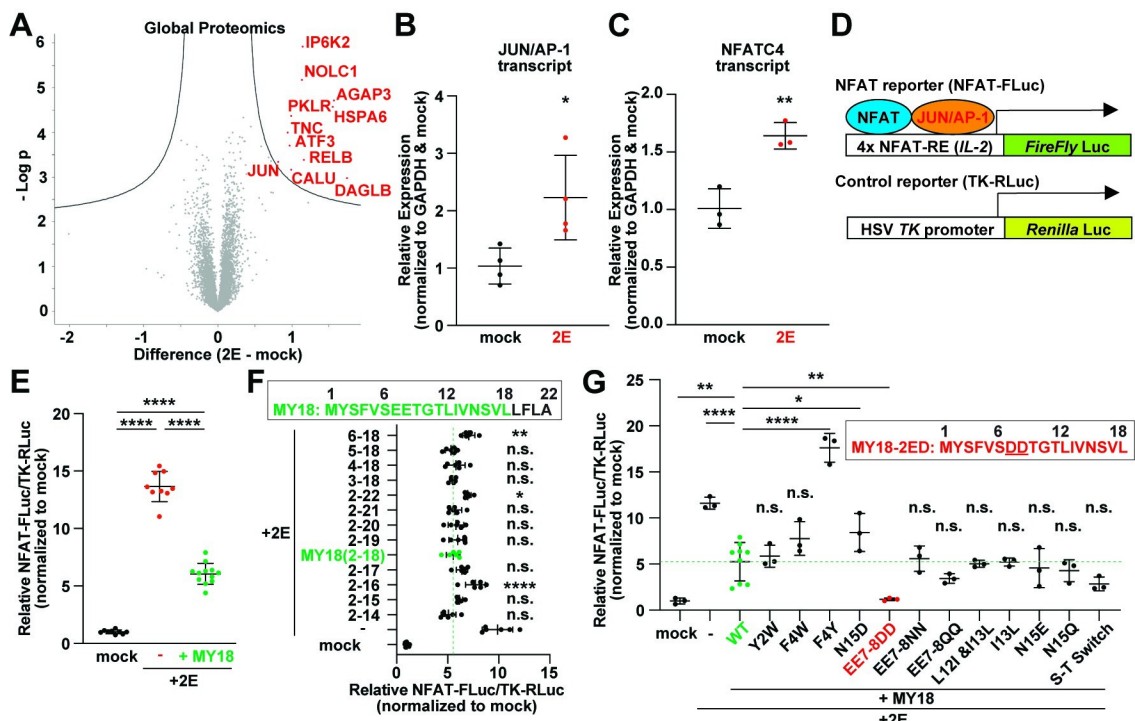

**Fig 2. Mutagenesis of MY18 peptide to develop iPep-SARS2-E.** (**A**) Volcano plots of global proteomics of HEK 293S cells transfected with SARS2-E fused to mKate2 (2E-mKate2). Red plots demonstrate significant increases in 2E-mKate2 compared to empty vector (mock) (FDR: 0.05). (**B** and **C**) The expressions of *JUN*/AP-1 (**B**) and *NFATC4* transcripts (**C**) significantly increased in HEK 293S cells transfected to 2E-mKate2 compared to mock. Unpaired Student's *t* test was used (** *P* < 0.01; * *P* < 0.05, *n* = 3–4). (**D**) Schematic representation of dual luminescence reporter system using 4-repeated NFAT response element (RE) of human *IL-2* gene and *Firefly* luciferase gene (NFAT-FLuc) and herpes simplex virus thymidine kinase (HSV *TK*) promoter-driven *Renilla* luciferase gene (TK-RLuc) as transfection control. (**E**) Relative NFAT-FLuc/TK-RLuc activity in HEK 293T cells transfected using empty vector (mock, *n* = 9) or 2E-mKate2 plasmid without (-, *n* = 9) and with MY18 plasmid (+MY18, *n* = 12). One-way ANOVA with Tukey's multiple comparisons test was used (**** *P* < 0.0001). (**F**) Relative NFAT-FLuc/TK-RLuc activity in HEK 293T cells transfected using empty vector (mock, *n* = 9) or 2E-mKate2 plasmid without (-, *n* = 6) and with various sizes of SARS2-E amino-terminal constructs including MY18 (each, *n* = 6, inset). One-way ANOVA with Dunnett's multiple comparisons test was used (**** *P* < 0.0001; ** *P* < 0.01; * *P* < 0.05; n.s., not significant, compared to MY18). (**G**) Relative NFAT-FLuc/TK-RLuc activity in HEK 293T cells transfected using empty vector (mock, *n* = 3) or 2E-mKate2 plasmid without (-, *n* = 3) and with MY18 mutants (each, *n* = 3). One-way ANOVA with Dunnett's multiple comparisons test was used (**** *P* < 0.0001; ** *P* < 0.01; * *P* < 0.05; n.s., not significant, compared to MY18 wild-type, WT, *n* = 9). S-T switch, Ser and Thr replaced each other. The inset demonstrates the amino acid sequence of MY18-2ED (EE7-8DD, underlined) mutant, which significantly improves the inhibitory effect on SARS2-E-mediated NFAT-JUN/AP-1 activation. The data underlying this figure can be found in S1 Data. All the graphs in the figure are mean ± SD. FDR, false discovery rate; HEK, human embryonic kidney.

confirmed that iPep-SARS2-E had the same rescuing effect on lysosomal pH phenotype in 2E-transfected mammalian cells as the plasmid construct did (S2A Fig).

To characterize iPep-SARS2-E, we first produced a monoclonal antibody against the N-terminal region of 2E. Following previous studies of viral envelope topology, we believed that the N-terminus may be in the extracellular region of SARS-CoV-2 [25,36]. To produce the monoclonal antibody, we used a synthetic peptide composed of the first 18 amino acids of E protein (i.e., MY18) with keyhole limpet haemocyanin (KLH) as the antigen. Western blotting result reveals that a hybridoma produces a 2E monoclonal antibody (2E-N; clone, N2A5E8) that can recognize 2E proteins expressed in a mammalian heterologous expression system (S2B Fig). Next, we conducted an ELISA using the antibody to compare the affinity of our 2E-N antibody to our 2ED and wild-type peptides. We confirmed that the 2ED mutation reduces the binding affinity of the 2E antibody, which was produced using the wild-type MY18 peptide as the

antigen (S2C Fig). Next, we used ELISA to examine how stable iPep-SARS2-E is in phosphate-buffered saline (PBS) at 37˚C. We did not observe obvious peptide degradation in 24 h, though the peptide might become unstable after 48 h because the standard deviation became larger compared to earlier time points (S2D Fig). Using an apoptosis/necrosis assay with flow cytometry, we confirmed that iPep-SARS2-E does not cause cellular toxicity in mammalian cells in vitro (S2E–S2G Fig).

To investigate the interaction of MY18-2ED and 2E protein biochemically, we first conducted immunoprecipitation using 6xHis-tagged MY18-2ED (His-MY18) and 2E-YFP with Ni column. The western blotting using anti-GFP antibody, which can recognize 2E-YFP protein band, demonstrates that Ni column with His-MY18 can pull down 2E-YFP proteins (Fig 3A), suggesting an interaction between MY18-2ED and 2E protein, though there was no obvious difference in MY18-2E protein interaction between the wild-type and 2ED peptide constructs, according to the immunoprecipitation result (S2H Fig). To confirm this in situ, we co-expressed 2E-mKate2 with His-MY18 in NIH 3T3 cells using lipofection and anti-His tag antibody conjugated to Alexa Fluor 488 to examine whether MY18 peptide interacts 2E protein directly. The fluorescent imaging result reveals that His-MY18 are co-localized with 2E proteins in the mammalian cells (Fig 3B). To further investigate the molecular mechanism underlying the inhibitory effect of MY18-2ED, we next used our established electrophysiological recording [19] to examine whether MY18-2ED co-transfected with 2E has a direct effect on 2E channel activity. The result demonstrates that MY18-2ED significantly reduced 2E channel current in HEK cells (Fig 3C and 3D), suggesting that MY18-2ED may inhibit 2E function directly. Together, the results suggest that MY18-2ED might be integrated into 2E protein oligomers and inhibit 2E function.

To examine the effect of MY18-2ED on 2E protein, we incubated mammalian cells with iPep-SARS2-E and transfected them with 2E-YFP construct. Interestingly, iPep-SARS2-E significantly reduced 2E-YFP protein expression in mammalian cells. While we observed both monomeric and aggregate bands of 2E proteins, both forms were significantly decreased in the treated cells (Figs 3E–3G and S3A). As a control experiment, we used YFP-transfected cells with peptide treatment and did not observe any effect of iPep-SARS2-E on YFP protein expression (Figs 3H, S3B and S3C). The results indicate that iPep-SARS2-E does not affect lipofection but reduces 2E expression in mammalian cells. Based on the results obtained in the series of experiments, the suggested molecular mechanism underlying the inhibitory effect of iPep-SARS2-E is as follows (Fig 3I): the peptides might interact with 2E proteins (Fig 3A and 3B) and inhibit 2E channel function (Fig 3C and 3D), resulting in lysosomal pH restoration (Figs 1B–1E and S2A) and 2E protein degradation (Fig 3E–3G).

To examine the kinetics of peptide penetration into mammalian cells, we conjugated a fluorescent probe, Alexa Fluor 594, to the N- or C-terminus of iPep-SARS2-E. Fluorescent imaging in situ reveals that the C-terminal fused version exhibited faster cell-penetration than the N-terminal version in NIH 3T3 cells. We found that most cells can uptake the peptides in 2 h (S4A and S4B Fig). This result suggests that the N-terminal conjugation of Alexa Fluor 594 might slightly affect the function of the TAT motif in the peptide. To examine the off kinetics, we next treated cells with iPep-SARS2-E fluorescent peptide (C-terminal version, TAT-MY18-2ED-Alexa Fluor 594) for 24 h, washed out the culture medium, and then monitored the red fluorescence (S4C Fig). The in situ imaging suggests that the peptide is not reduced or degraded for 96 h; however, the peptide might become unstable and aggregated after 72 h at 37˚C in the live cells, since larger fluorescent puncta were observed compared to earlier time points (S4D Fig). In addition, we tested the C-terminal version in another mammalian cell line, Vero-E6, which has been commonly used in virological studies using SARS-CoV-2. We confirmed that the peptides can penetrate into these cells as well (S5A and S5B Fig).

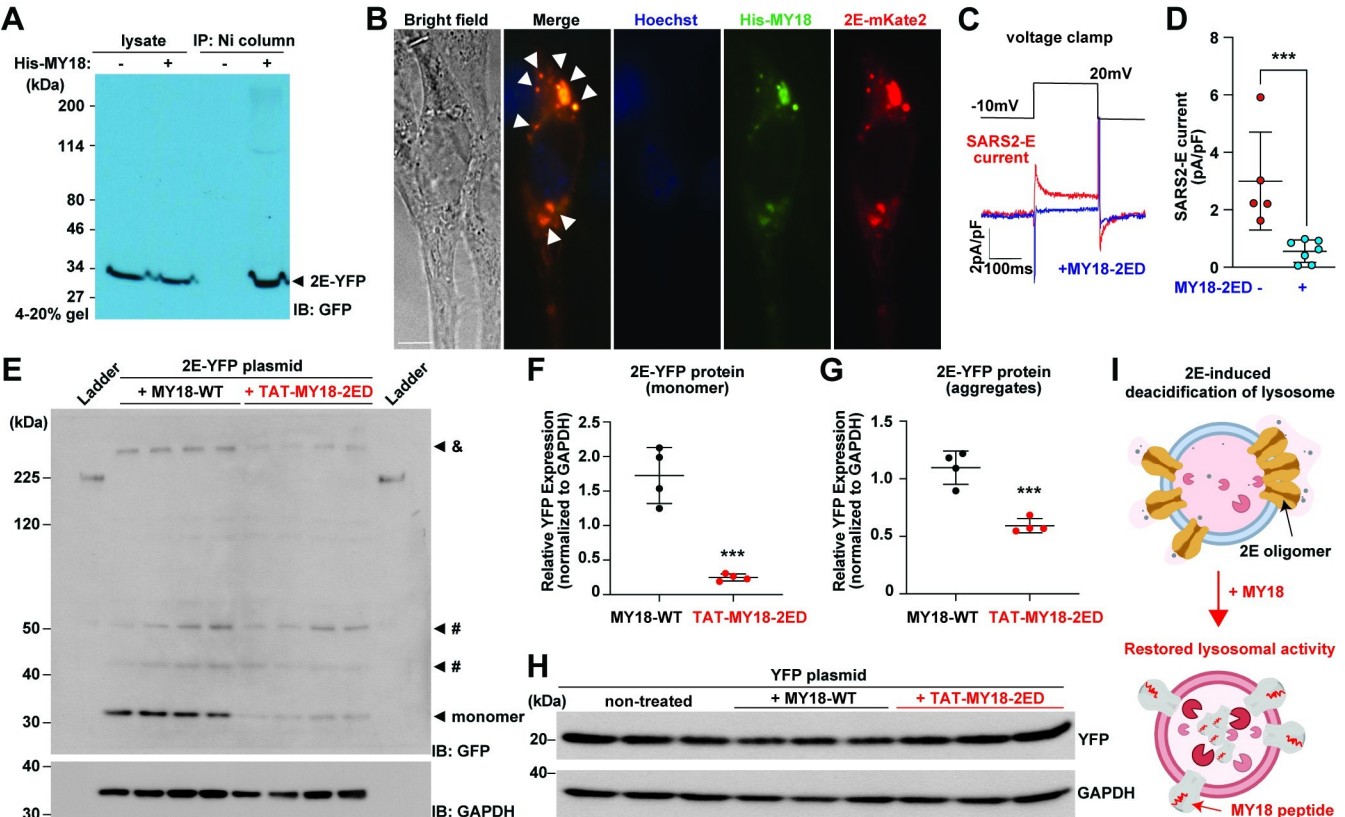

**Fig 3. Characterization of iPep-SARS2-E.** (**A**) Immunoprecipitation of 2E protein using Ni column and HEK 293T cells transfected using 2E-YFP with or without 6xHis-MY18-2ED (His-MY18). Anti-GFP antibody was used to blot 2E-YFP protein bands. (**B**) Representative epi-fluorescence image of NIH 3T3 cells co-transfected with 2E-mKate2 (red) and His-MY18 plasmids. Anti-His tag antibody conjugated to Alexa Fluor 488 (green) and Hoechst 33258 dye (blue, nucleus) were used after cell fixation. White arrowheads, co-localization of red and green fluorescence. Scale bar, 5 μm. (**C**) Representative SARS2-E currents in 2E-PM-expressing HEK 293S cells mock-transfect and co-transfected with MY18-2ED peptide construct. (**D**) MY18-2ED significantly blocked 2E currents. Student's $t$ test was used (*** $P < 0.001$). (**E**) Representative immunoblot images of HEK 293T cells transfected with SARS2-E fused with YFP (2E-YFP) and 48 h treated with 10 μM TAT-MY18-2ED or MY18-WT peptides (negative control). Anti-GFP (for 2E-YFP, top) and GAPDH antibodies (as loading control, bottom) were used. Putative aggregates of 2E-YFP proteins (&) were observed even though Urea-based lysis buffer was used for the sample preparation. #, nonspecific bands around 40 and 50 kDa according to the YFP blotting (S3B Fig). The whole blotting image of GAPDH is shown in S3A Fig. (**F and G**) Quantification of 2E-YFP monomeric form (**F**) and aggregates (**G**) of HEK 293T cells transfected with 2E-YFP and treated using TAT-MY18-2ED ($n = 4$) or MY18-WT peptides ($n = 4$). Student's $t$ test was used (*** $P < 0.001$). (**H**) Representative immunoblot images of HEK 293T cells transfected with YFP plasmid and 48 h treated with TAT-MY18-2ED (10 μM, $n = 3$) and MY18-WT peptides (10 μM, $n = 3$, a negative control) and non-treated cells ($n = 3$, another negative control). Anti-GFP (for YFP, top) and GAPDH antibodies (as loading control, bottom) were used. The whole blotting images and quantification are shown in S3B and S3C Fig, respectively. (**I**) Schematic representation of the molecular mechanism underlying the effect of MY18 peptide on 2E protein and lysosomal function. The images are prepared using BioRender: Top, 2E induces deacidification in lysosome (Fig 1B and 1C); bottom, MY18 peptides binds 2E proteins (Fig 3A and 3B), resulting in 2E inhibition (Fig 3C and 3D) and restored lysosomal activity (Figs 1B, 1C and S2A) and 2E protein reduction (Fig 3E–3G). The data underlying this figure can be found in S1 Data. All the graphs in the figure are mean ± SD. HEK, human embryonic kidney; NIH, National Institute of Health.

To examine the effect of iPep-SARS2-E on SARS-CoV-2 infection, as a proof-of-concept experiment in vitro, we conducted a cytopathic assay using a mammalian cell line, Vero-E6, and SARS-CoV-2 (WA1 strain, Fig 4A). We used the wild-type MY18 peptide (non-TAT version, MY18-WT) as a negative control in the assay because MY18-WT does not have any cell-penetrating motifs or effect on 2E in the pH imaging in NIH 3T3 cells (Fig 1E). The cytopathic assay result demonstrates that iPep-SARS2-E significantly inhibits viral toxicity in vitro (Fig 4B: IC$_{50}$ of iPep-SARS2-E, approximately 400 nM). Following these results, we next conducted time-course experiments to elucidate the mechanism underlying the inhibitory effect of iPep-SARS2-E on viral function (Fig 4C). The qPCR result demonstrates that there is no difference

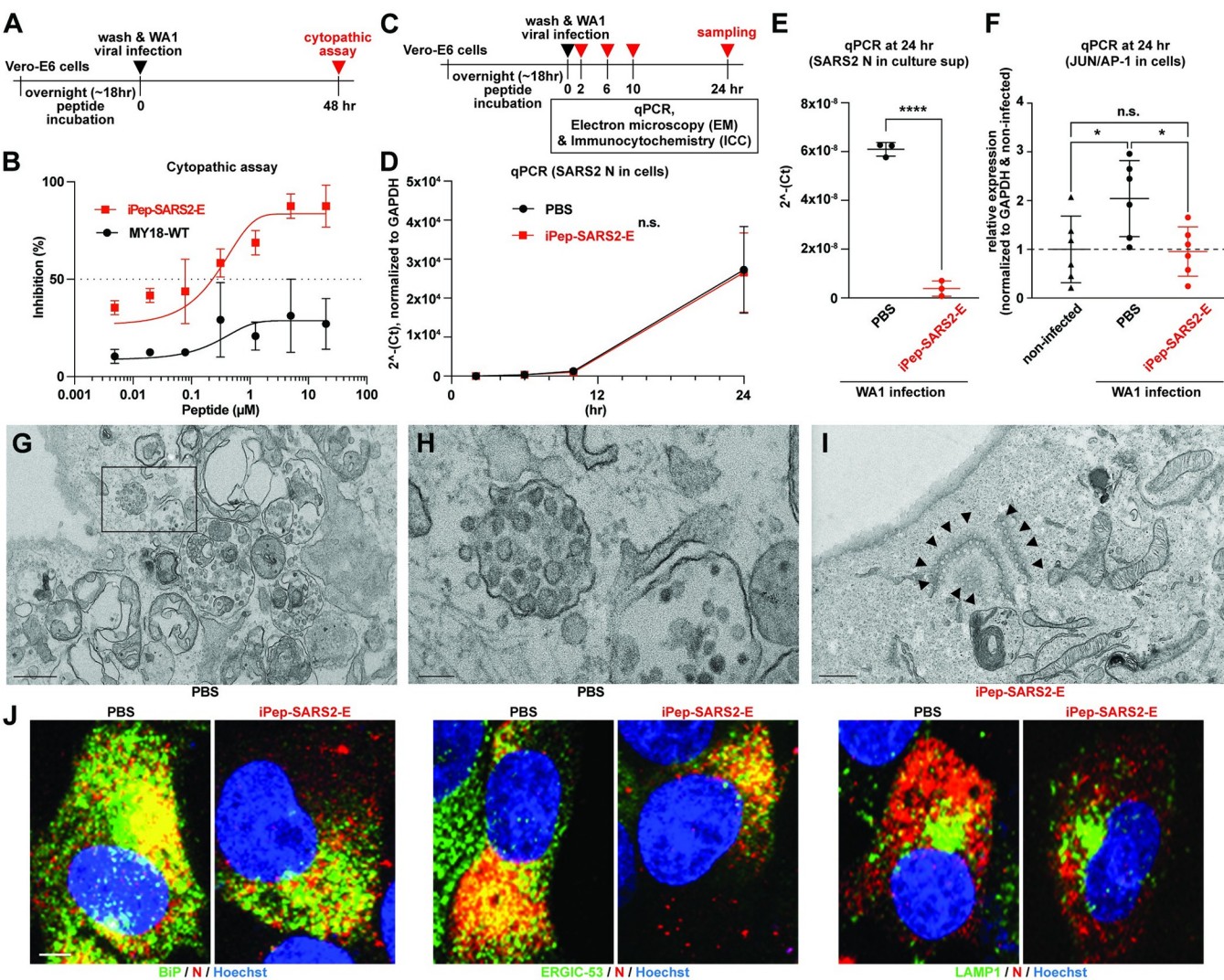

**Fig 4. iPep-SARS2-E in vitro test.** (**A**) Experimental design for the iPep-SARS2-E (TAT-MY18-2ED) test using SARS-CoV-2 WA1 virus (MOI, 0.10) and Vero-E6 cells in vitro. (**B**) Inhibition of iPep-SARS2-E in the cytopathic effect of SARS-CoV-2 WA1 virus on Vero-E6 cells. MY18-WT (non-TAT) was used as a control. (**C**) Design of time-course experiment using SARS-CoV-2 WA1 virus (MOI, 0.10) and Vero-E6 cells in vitro. (**D**) Time-course qPCR of SARS2-CoV-2 nucleocapsid (SARS2 N) expression of PBS- and iPep-SARS2-E (10 μM)-treated Vero-E6 cells. The expression of N gene was normalized to a house-keeping gene, GAPDH. Student's $t$ test was used at each time point (n.s., not significant). (**E**) qPCR of SARS2 N expression of PBS- and iPep-SARS2-E (10 μM)-treated Vero-E6 cell culture supernatant (sup) at 24-h post-infection. Student's $t$ test was used (**** $P < 0.0001$). (**F**) qPCR of JUN/AP-1 expression of PBS- and iPep-SARS2-E (10 μM)-treated Vero-E6 cells comparing to non-infected cells. The expression of JUN was normalized to GAPDH. One-way ANOVA with Tukey's multiple comparisons test was used (* $P < 0.05$; n.s., not significant). (**G**) Representative electron microscopic (EM) image of PBS-treated Vero-E6 cells at 24 h post-infection. Scale bar, 500 nm. (**H**) Higher magnification of the EM image of PBS-treated Vero-E6 cells at 24 h post-infection (a box shown in Fig 4G). Scale bar, 100 nm. (**I**) Representative EM image of iPep-SARS2-E-treated Vero-E6 cells at 24 h post-infection. Arrowheads, virus-like particles accumulated in the endoplasmic reticulum. The other EM image is also shown in S6A Fig. Scale bar, 500 nm. (**J**) Representative confocal fluorescent images of Vero-E6 cells treated with PBS or iPep-SARS2-E at 24 h post-infection. SARS2 N antibody (red) and Hoechst 33258 dye (blue, for nucleus) were used with antibodies of subcellular organelle markers (green): BiP for endoplasmic reticulum (ER), ERGIC-53 for ER Golgi inter compartment (ERGIC), and LAMP1 for lysosome. Scale bar, 5 μm. The data underlying this figure can be found in S1 Data. All the graphs in the figure are mean ± SD. EM, electron microscopic; ER, endoplasmic reticulum; ERGIC, ER-Golgi inter-compartment; PBS, phosphate-buffered saline; qPCR, quantitative polymerase chain reaction; SARS-CoV-2, Severe Acute Respiratory Syndrome Coronavirus 2.

in SARS-CoV-2 nucleocapsid (N) gene expression, suggesting no effect of iPep-SARS2-E on virus transcription and entry (Fig 4D). On the other hand, there is a significant difference in SARS-CoV-2 N gene detection between the culture supernatants of PBS-treated control and iPep-SARS2-E-treated cells sampled at 24 h post-infection (Fig 4E), demonstrating a

significant reduction of virus release from iPep-SARS2-E-treated cells. Importantly, we found that iPep-SARS2-E could significantly restore the expression of JUN/AP-1 (Fig 4F), which we had used as a reporter of SARS2-E cellular toxicity in this study to optimize the MY18 peptide series (Fig 2). Electron microscopy reveals that viral particles can be observed in large vacuoles of PBS-treated cells (Fig 4G and 4H), which could be deacidified and disrupted lysosomes, according to a previous study [37]. However, no vacuoles containing multiple viral particles were found in iPep-SARS2-E-treated cells, while small particles were found in the endoplasmic reticulum and nuclear envelope (Figs 4I and S6A). This suggests that the particles might be virions, though it is not clear whether the virions are mature in iPep-SARS2-E-treated cells. Therefore, we next examined infectivity of these intracellular particles from iPep-SARS2-E-treated cells treated via an endpoint titration assay used to quantitate intracellular virus particles (S6B Fig) [38–40]. We found that there was a modest but significant ($P < 0.0001$) reduction in virus titers of iPep-SARS2-E-treated cell samples compared to PBS-treated control cells, but that the intracellular particles of iPep-SARS2-E-treated cell samples are still infectious (S6C Fig).

E protein is thought to be involved in Spike protein protection and maturation [20,24]. Therefore, we examined the effect of iPep-SARS2-E on Spike protein expression. The western blotting result demonstrates significant reductions of Spike protein expression (S6D Fig). Immunocytochemistry confirms that, in iPep-SARS2-E-treated cells, N proteins are co-localized to BiP/GRP78 (a marker of endoplasmic reticulum, or ER), and to ERGIC-53 (a marker of ER-Golgi inter-compartment, or ERGIC); in PBS-treated cells, N proteins were more highly and broadly expressed, and co-localized to LAMP1 (a lysosomal marker) (Fig 4J).

Following these results, we conducted in vitro experiments using iPep-SARS2-E at a later time-point to examine its effect further. qPCR shows that there is a significant decrease in SARS-CoV-2 N and E gene expression in iPep-SARS2-E-treated Vero-E6 cells at 48 h post-infection compared to the PBS-treated control cells (S6E and S6F Fig). In addition, iPep-SARS2-E significantly restores the expression of JUN/AP-1 (S6G Fig). Importantly, detection of SARS-CoV-2 genes in the culture supernatant harvested at 48 h post-infection is significantly reduced in iPep-SARS2-E-treated cells compared to PBS-treated control cells (S6H Fig), demonstrating that iPep-SARS2-E suppresses virus release. Immunocytochemistry shows that the majority of infected Vero-E6 cells became round and apoptotic-like in the PBS-treated group, while iPep-SARS2-E-treated cells exhibited moderate expression of viral N and normal cellular morphology (S6I Fig). This is consistent with the results of the cytopathic assay (Fig 4B).

Next, to validate the inhibitory effect of iPep-SARS2-E further, we conducted a preclinical experiment in vivo using iPep-SARS2-E intravenous (i.v.) injection to Balb/c mice infected with a mouse-adapted strain of SARS-CoV-2 (MA10, [41–43]). First, we conducted i.v. injection of the C-terminal fluorescent version of iPep-SARS2-E (TAT-MY18-2ED-Alexa Fluor 594) to mice and confirmed that the peptide can permeate and is detectable in mouse lung tissues 2 h after administration (Fig 5A). Following this result, we next injected iPep-SARS2-E post-infection. The mice were sacrificed 4 days after MA10 viral infection, and their lungs were harvested for viral titer, qPCR, and histology (Fig 5B). The results show no difference in body weight between the groups, but a significant reduction of viral propagation in iPep-SARS2-E-treated mouse lungs compared to the control (Fig 5C and 5D). qPCR also confirms the inhibitory effect of iPep-SARS2-E on the viral propagation in vivo (Fig 5E). Lung histology reveals no immune infiltration or alveolar damage in iPep-SARS2-E-treated mouse lungs, while minimal interstitial infiltrates with patchy lymphoid aggregates and protein accumulation were observed in the non-treated control group (Fig 5F).

To examine whether iPep-SARS2-E could be applied for prevention of SARS-CoV-2 infection, we conducted another experimental series in vivo using iPep-SARS2-E intranasal

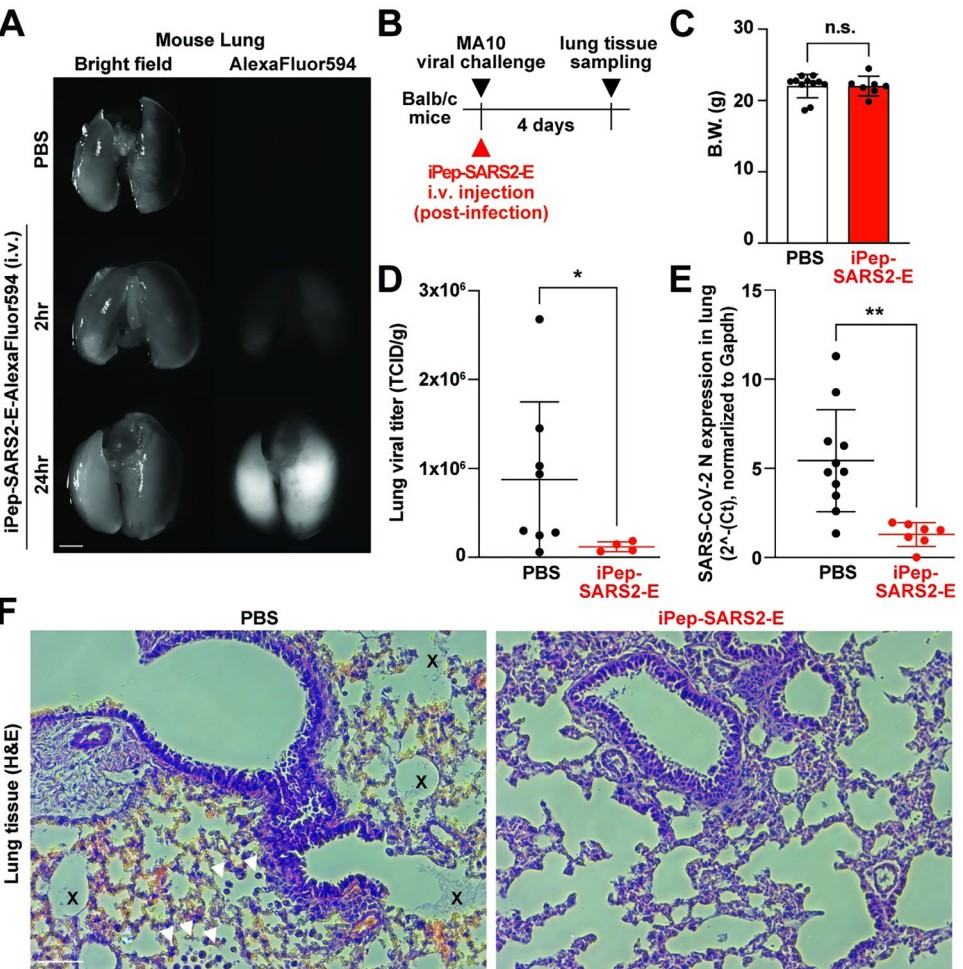

**Fig 5. iPep-SARS2-E in vivo preclinical study.** (**A**) Representative fluorescent and bright field images of lung tissues isolated from mice administrated i.v. with PBS or Alexa594-conjugated iPep-SARS2-E peptide (TAT-MY18-2ED-A594, 300 μM, 2 h and 24 h). After isolating the tissues, the samples were washing using PBS 3 times, and the fluorescent and bright field images were taken by a fluorescent stereoscope. Scale bar, 2 mm. (**B**) Experimental design using the iPep-SARS2-E i.v. injection in vivo. (**C**) There is no difference in body weight between PBS control ($n = 11$) and iPep-SARS2-E-treated mice ($n = 7$). (**D**) There is a significant reduction of lung viral titer in iPep-SARS2-E-treated mice compared to the control. Student's $t$ test was used (* $P < 0.05$; n.s., not significant). Median TCID is normalized to lung wet weight (g) measured before tissue homogenization to isolate the virus. (**E**) iPep-SARS2-E significantly reduced the transcript expression of SARS2 N in MA10-infected Balb/c mouse lung tissues (PBS, $n = 11$; iPep-SARS2-E, $n = 7$, normalized to mouse Gapdh expression). Student's $t$ test was used (** $P < 0.01$). (**F**) Representative images of hematoxylin and eosin (HE) staining of mouse lung tissues of PBS control and iPep-SARS2-E-treated mice at 4 days post-infection. Protein accumulation (x) and immune cells (arrowheads) are indicated. Scale bar, 50 μm. The data underlying this figure can be found in S1 Data. All the graphs in the figure are mean ± SD. HE, hematoxylin and eosin; PBS, phosphate-buffered saline; TCID, tissue culture infection dose.

administration to Balb/c mice infected with MA10 virus. First, we administrated the C-terminal fluorescent version of iPep-SARS2-E to mice intranasally and confirmed that the peptide can permeate and is detectable in mouse nasal tissues 2 h after administration (S7A and S7B Fig). Next, we conducted a safety study in vivo (S7C Fig) using intranasal administration to Balb/c mice to examine the effect of iPep-SARS2-E on body weight and inflammation markers, comparing to non-treated and PBS-administrated groups. We did not find any significant differences in body weight, Cxcl12 and C5a among iPep-SARS2-E-, PBS-, and non-treated groups (S7D–S7F Fig). Following the results using intranasal administration, we applied iPep-

SARS2-E intranasal administration to Balb/c mice infected with MA10 virus (S7G Fig). We found that iPep-SARS2-E significantly prevents the body weight loss and suppresses 2E protein expression in infected mouse lungs in vivo (S7H–S7K Fig). These results using in vivo experiments demonstrate that SARS2-E inhibition can be a novel strategy to prevent SARS-CoV-2 toxicity and propagation in vivo.

To validate iPep-SARS2-E further, we continued to conduct control experiments, preparing an additional negative control and experimental conditions. When we examined the effect of deletion on MY18 constructs using the luciferase reporter, we had found that none of the constructs significantly improved the effect: the majority did not reduce its efficacy significantly, but deletion at the 5th and 17th/18th residues significantly reduced the inhibitory effect of MY18 on 2E-mediated NFAT/AP-1 pathway alteration (Fig 2F). Following the results, we introduced Ala-substitution, targeting these amino acids of MY18 (S8A Fig). NFAT/AP-1 Luc assay and DND-189 imaging results demonstrate that the mutagenesis significantly reduced the inhibitory effect of MY18, although NFAT/AP-1 Luc assay result suggests that it was not sufficient to fully counteract the effect of MY18 (S8B and S8C Fig). After further mutagenesis, we found that the addition of an L12A substitution was able to negate the inhibitory effect of MY18 against 2E. We confirm using qPCR, that, like PBS, the mutant peptide cannot reduce SARS-CoV-2 N expression in Vero-E6 cell culture supernatant (S8D Fig). In the following experiments, we used this mutant peptide as a new negative control.

The qPCR results using Vero-E6 cell culture supernatant and lysate samples at 24 h post-infection with SARS-CoV-2 (Fig 4C–4E) suggest that iPep-SARS2-E may not have any effect on the virus entry. However, this time-course transcription profiling may not be sufficient as a readout of the virus entry. To address this concern, we conducted an experiment using pseudo virus containing SARS-CoV-2 Spike, Membrane, E proteins, and YFP reporter with the negative control peptide and iPep-SARS2-E because pseudo virus is useful to examine the effect of drug candidates on the virus entry [44]. The pseudo virus infection resulted in no difference in YFP-positive cell number between the negative control peptide- and iPep-SARS2-E-treated cells, revealing no effect of iPep-SARS2-E on the virus entry (S8E and S8F Fig).

To validate the inhibitory effect of iPep-SARS2-E peptide further, we conducted in vitro experiments using human pluripotent stem cell-derived branching lung organoids and WA1 virus (Fig 6A and 6B). We found that there is a significant reduction in SARS-CoV-2 N transcript of the culture supernatant at 24 h post-infection between the negative-control peptide- and iPep-SARS2-E-treated organoids but not in the organoid lysate (Fig 6C and 6D), suggesting that virus egress is blocked by iPep-SARS2-E. Immunocytochemistry allows us to observe higher expression of SARS-CoV-2 N in the negative control peptide-treated organoids compared to iPep-SARS2-E-treated organoids (Fig 6E). The results using the lung organoids are consistent with the results using Vero-E6 cells at 24 h post-infection (Fig 4). Our results in monolayer cells and in human 3D lung organoids reveal a new strategy to prevent SARS-CoV-2 propagation using iPep-SARS2-E, which inhibits 2E activity and virus egress.

Next, we conducted another in vivo mouse study (S8G Fig) using a single intranasal administration to Balb/c mice to examine the effect of iPep-SARS2-E and the negative control peptide on body weight, SARS-CoV-2 titer, and N transcript in the mouse lungs. We found that iPep-SARS2-E could significantly prevent their body weight loss while the negative control did not (S8H Fig). The viral titer and qPCR results confirm the inhibitory effect of iPep-SARS2-E in vivo (S8I and S8J Fig). These in vivo results demonstrate that SARS2-E inhibition using a single intranasal dose of iPep-SARS2-E can be useful to prevent SARS-CoV-2 toxicity and propagation in vivo.

Next, we hypothesized that this peptide design and strategy could be customized and applied to other human coronaviruses because coronavirus E proteins are highly conserved

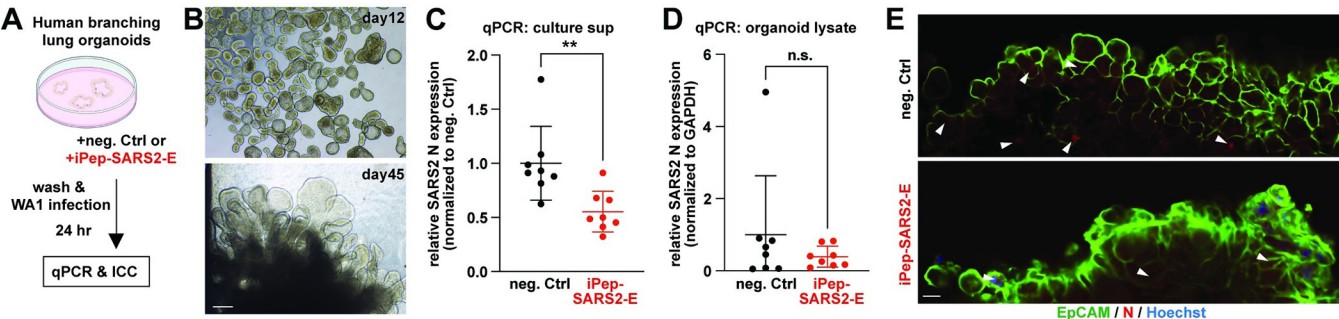

**Fig 6. iPep-SARS2-E and its negative control peptide test using in vitro human lung organoid model.** (**A**) Experimental design for the iPep-SARS2-E test using SARS-CoV-2 WA1 virus (MOI, 0.10) and human pluripotent stem cell-derived branching lung organoids in vitro. The image is prepared using BioRender. (**B**) Representative phase contrast images of human pluripotent stem cell-derived branching lung organoids. Scale bar, 50 μm. (**C and D**) qPCR of SARS2 N expression of the negative control mutant peptide (neg. Ctrl, S8 Fig)- and iPep-SARS2-E (10 μM)-treated organoids culture supernatant (sup, **C**) and cells (**D**) at 24 h post-infection. Student's *t* test was used (* *P* < 0.05; n.s., not significant). (**E**) Representative merged section images of confocal fluorescence and bright field of human lung organoids treated with neg. Ctrl or iPep-SARS2-E at 24 h post-infection. SARS2 N antibody (red, arrowheads), EpCAM antibody (green), and Hoechst dye (blue, for nucleus) were used. Scale bar, 20 μm. The data underlying this figure can be found in S1 Data. All the graphs in the figure are mean ± SD. qPCR, quantitative polymerase chain reaction; SARS-CoV-2, Severe Acute Respiratory Syndrome Coronavirus 2.

(Figs 7A and S9). Therefore, following iPep-SARS2-E development, we designed inhibitory peptide constructs against E proteins from each of the other human coronaviruses: MERS-CoV, HCoV-NL63, -OC43, -HKU1, and -229E (Fig 7B). First, we found that the overexpression of MERS-CoV and HCoV-NL63 E proteins significantly increased NFAT/AP-1 luciferase reporter activity in mammalian cells while the E proteins of HCoV-OC43, -HKU1, and -229E do not have an effect on NFAT/AP-1 pathway (Fig 7C). Following these results, we focused our testing to MER-CoV and HCoV-NL63 MY18 WT peptide constructs using the NFAT/AP-1 luciferase reporter assay. The reporter assay results demonstrate that MERS-CoV and HCoV-NL63 MY18 WT could significantly reduce the effect of each E protein on NFAT/AP-1 reporter while substitution of Glu/E to Asp/D or Asp/D to Glu/E does not improve the MY18 constructs for MERS-CoV and HCoV-NL63, respectively (Fig 7D and 7E). To improve each MY18 further, we conducted mutagenesis of MERS-CoV MY18 and HCoV-NL63 MY18 constructs. We found that HCoV-NL63 MY18 2DE and N9D and MERS-CoV MY18 R8H are the best to inhibit the effect of HCoV-NL63 and MERS E proteins on NFAT/AP-1 pathway in mammalian cells, respectively (Fig 7D and 7E). Next, using DND-189 pH imaging, we found that the overexpression of MERS-CoV, HCoV-NL63, and -HKU1 E proteins significantly reduced DND-189 fluorescence in mammalian cells while the E proteins of HCoV-OC43 and -229E do not have a significant effect on lysosomal proton homeostasis (Fig 7F). Following these results, we focused on testing MY18 constructs on MERS-CoV, HCoV-NL63, and -HKU1 and found that MERS-CoV MY18 R8H, HCoV-NL63 MY18 2DE and N9D, HCoV-HKU1 MY18 D8E peptide constructs could significantly rescue the phenotypes in proton homeostasis in mammalian cells caused by the respective E proteins (Fig 7F). The results of this experiment reveal that the MY18 peptides can be applicable for not only SARS-CoV-2 but also some of the other human coronaviruses such as MERS-CoV, HCoV-NL63, and HCoV-HKU1, demonstrating that E protein can more broadly be a potential therapeutic target for human coronaviruses.

## Discussion

We applied a pH fluorescent dye, DND-189, to identify TAT-MY18 as a therapeutic candidate against E proteins (Fig 1). However, uptake of the dye could be dependent on cell viability and

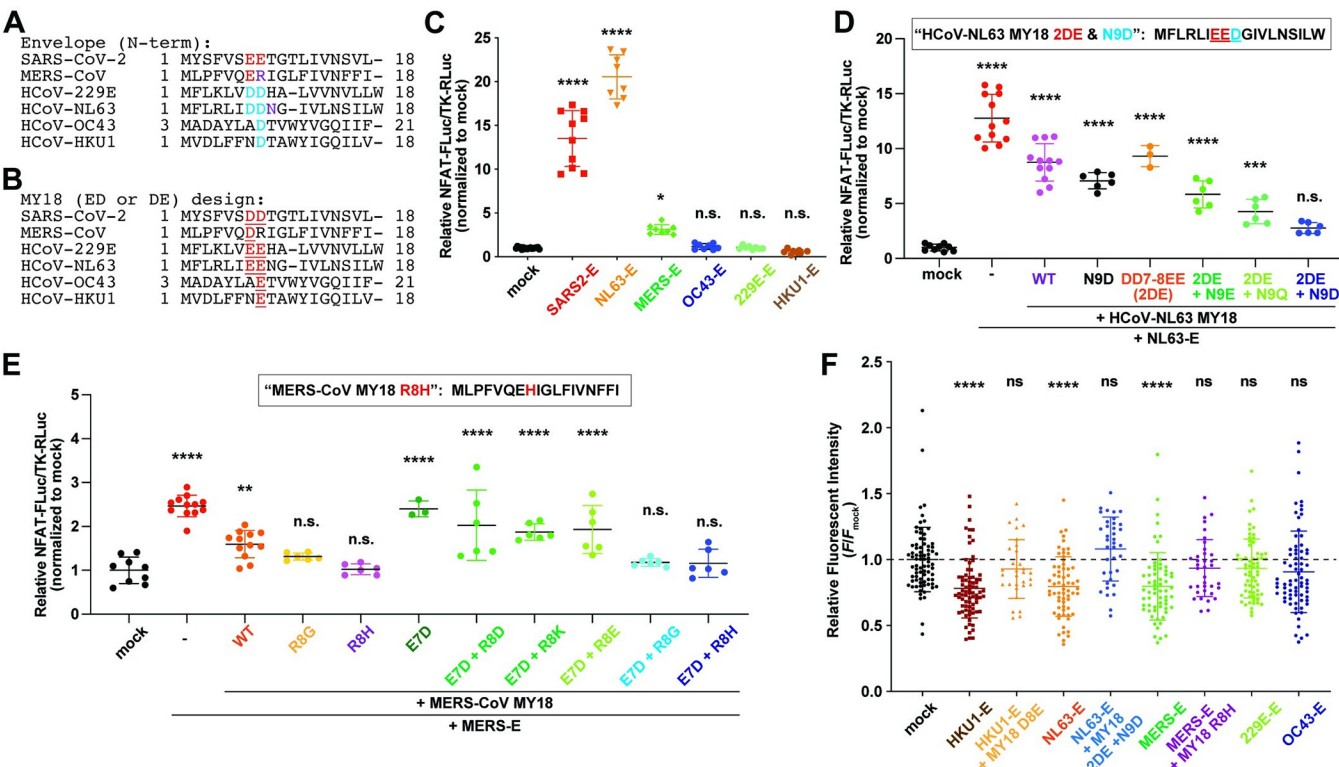

**Fig 7. MY18 peptide application for other human coronaviruses.** (**A**) Alignment of the N-terminal region of human coronavirus Envelope (E) of SARS-CoV-2, MERS-CoV, HCoV-229E, HCoV-NL63, HCoV-OC43, and HCoV-HKU1. Glu (red), Asp (blue), and other amino acid residues (purple) are targeted for further mutagenesis to customize MY18 inhibitory peptides for each viral E. (**B**) MY18 peptide design for each human coronavirus E of SARS-CoV-2, MERS-CoV, HCoV-229E, HCoV-NL63, HCoV-OC43, and HCoV-HKU1. Glu and Asp are replaced (red), following iPep-SARS2-E. (**C**) Relative NFAT-FLuc/TK-RLuc activity in HEK 293T cells transfected using mock ($n = 17$), SARS2-E ($n = 10$), and the other coronavirus E (each, $n = 8$). One-way ANOVA with Dunnett's multiple comparisons test was used (**** $P < 0.0001$; * $P < 0.05$; n.s., not significant, compared to mock). (**D**) Relative NFAT-FLuc/TK-RLuc activity in HEK 293T cells transfected using mock ($n = 9$), NL63-E without ($n = 12$), and with NL63-MY18 constructs (WT, $n = 12$; 2DE, $n = 3$; the other mutants, $n = 6$). One-way ANOVA with Dunnett's multiple comparisons test was used (**** $P < 0.0001$; *** $P < 0.001$; n.s., not significant, compared to mock). (**E**) Relative NFAT-FLuc/TK-RLuc activity in HEK 293T cells transfected using mock ($n = 9$), MERS-E without ($n = 12$), and with MERS-MY18 constructs (WT, $n = 12$; E7D, $n = 3$; the other mutants, $n = 6$). One-way ANOVA with Dunnett's multiple comparisons test was used (**** $P < 0.0001$; ** $P < 0.01$; n.s., not significant, compared to mock). (**F**) Relative fluorescent intensity of DND-189 dye in NIH 3T3 cells transfected with mock ($n = 78$) and the other human coronavirus E-mKate2 constructs: OC43 ($n = 74$), 229E ($n = 66$), MERS ($n = 63$), NL63 ($n = 65$), and HKU1 ($n = 78$). Each MY18 mutant construct was also tested: MERS-MY18 (R8H, $n = 36$), NL63-MY18 (2DE and N9D, $n = 37$), and HKU1-MY18 (D8E, $n = 31$). One-way ANOVA with Dunnett's multiple comparisons test was used (**** $P < 0.0001$; n.s., not significant, compared to mock). The data underlying this figure can be found in S1 Data. All the graphs in the figure are mean ± SD. HEK, human embryonic kidney; MERS-CoV, Middle East Respiratory Syndrome Coronavirus; SARS-CoV-2, Severe Acute Respiratory Syndrome Coronavirus 2.

intracellular organelle function because DND-189 is a single green fluorescent dye. Therefore, the fluorescent change might not be due to a direct effect of E protein, but a secondary effect. In addition to the DND-189 dye, we applied the NFAT/AP-1 luciferase reporter assay as a higher throughput screening platform to optimize the MY18 peptide using molecular biological approach (Fig 2). However, NFAT/AP-1 alteration might be a secondary consequence induced by other molecular phenotypes of E protein overexpression in mammalian cells. Therefore, immunoprecipitation, immunocytochemistry, and electrophysiological recording were useful to validate the inhibitory effect of the MY18 peptides on E protein function (Fig 3).

Control safety experiments in vitro and in vivo demonstrate that there is no obvious toxicity of iPep-SARS2-E in vitro and in vivo (S2E–S2G and S7C–S7F Figs). iPep-SARS2-E was shown to be efficacious against SARS-CoV-2 both in vitro and in vivo (Figs 4–6 and S6–S8). The

cytopathic assay result demonstrates that, MY18-WT, which was validated as a negative control in DND-189 imaging using NIH 3T3 cells (Fig 1E), also had a moderate inhibitory effect on the virus in Vero-E6 cells (Fig 4B), although no cell-penetrating motif is present. This suggests that the non-TAT version might be spontaneously taken up into Vero-E6 cells by endocytosis. Alternatively, we generated and validated another negative control peptide using mutagenesis (Figs 6 and S8).

iPep-SARS2-E significantly reduces 2E-YFP protein expression in mammalian cells while having no effect on YFP and GAPDH protein (Figs 3E–3H and S3A–S3C). Altered lysosomal pH due to 2E may result in a protective effect that allows for viral proteins to be sequestered from proteolysis and promote viral assembly and release as reported in the other viruses [19,37]. Therefore, it is possible that iPep-SARS2-E-induced 2E inhibition may restore lysosomal pH, allowing lysosomes to resume normal activity and resulting in proteolysis of 2E and Spike proteins (Figs 3E–3G and S6D). Importantly, the in vitro study suggests that iPep-SARS2-E may inhibit virus egress, since the viral transcript detection was significantly lower in iPep-SARS2-E-treated cell culture supernatant than in the PBS-treated control (Fig 4E). On the other hand, we found that there is no difference in SARS-CoV-2 N transcript expression and pseudo-virus infection between iPep-SARS2-E and the controls (Figs 4D, S8E and S8F), suggesting no effect of iPep-SARS2-E on virus entry and transcription. The cytopathic assay at later time-point suggests that the intracellular particles are still infectious in iPep-SARS2-E-treated cells (S6B and S6C Fig), though the stability of iPep-SARS2-E remains unclear over 48 h (S2D and S4D Fig). The western blotting result demonstrates that iPep-SARS2-E significantly reduced Spike protein expression in WA1-infected cells (S6D Fig). The results are consistent with the previous study reporting that E protein is involved in Spike protein expression and protection [20,24]. Together, these results suggest the molecular mechanism underlying the inhibitory effect of iPep-SARS2-E as follows: the peptides interact with 2E proteins and inhibit 2E channel function, resulting in lysosomal pH restoration and reduction of 2E- protective effect, which is crucial for the viral proteins, such as Spike, to be sequestered from proteolysis and promote virus egress.

E protein has been considered as a pharmaceutical target [45–50]. Compared to small molecules inhibiting E protein, iPep-SARS2-E might be more customizable to each human coronavirus E protein (Fig 7) and their future variants because it is peptide-based, though further optimization of MY18 peptides is required for each variant. The difference of this amino acid residue size (i.e., Glu versus Asp) might be crucial for human coronavirus E oligomerization. 2DE mutation could be applied for HCoV-NL63 version with N9D while ED mutation has a negative effect on the MERS-CoV version of MY18 peptide (Fig 7D and 7E). In our previous study for a bacterial lactate-binding protein LldR, the E103D mutant significantly increases the affinity and specificity of LldR to lactate [51]. Though E is a different molecule from LldR, E/D and D/E substitutions could be a useful strategy for peptide and protein engineering.

In summary, we established and applied 2 screening platforms using the lysosomal pH fluorescent imaging and the NFAT/AP-1 luminescence reporter system with our expertise in proteomics, imaging and human cellular modeling [52–54], to identify novel therapeutic candidates against human coronavirus E protein. Our inhibitory peptides can be customizable and applicable for targeting E protein of not only SARS-CoV-2 but also the other coronaviruses, such as MERS-CoV, HCoV-NL63, and HCoV-HKU1, demonstrating more broadly that E protein is a potential therapeutic target for human coronaviruses. Due to the unique effects on virus egress, iPep-SARS2-E could prove to be a promising therapeutic candidate against SARS-CoV-2, particularly if combined with other therapeutic candidates such as protease and polymerase inhibitors for a synergistic effect.

## Methods

### Ethical statement

Columbia University is a National Institute of Health (NIH) Office of Laboratory Welfare assured institution (ID#, D16-00003/A3007-01), which has complied with the NIH Public Health Service Policy and adhered to the standards in the guide for the care and use of laboratory animals. The animal study is approved by Columbia University Institutional Animal Care and Use Committee (protocol #, AC-AABP2571). This virus study using SARS-CoV-2 is approved by Columbia University ADARC committee and BSL3 facility committee.

### Cell culture

Human embryonic kidney (HEK) 293S cells (ATCC, Cat#CRL-3022) were cultured in Dulbecco's Modified Eagle Media Nutrient Mixture F-12 (DMEM/F-12, Thermo Fisher Gibco # 11320033) and HEK 293T cells (ATCC, Cat#CRL-3216), NIH 3T3 cells (ATCC, Cat#CRL-1658), and Vero-E6 cells (Cat# CRL-1586) were cultured in and DMEM (Thermo Fisher/Gibco #10313021). Both media were supplemented with GlutaMax-I and penicillin/streptomycin (PS) and 10% fetal bovine serum (FBS, not heat-inactivated, HyClone, #SH30071.03, Thermo Fisher) under normoxia (20% $O_2$, 5% $CO_2$, at 37˚C). The cell lines were passaged using trypsin-EDTA (0.25%, Thermo Fisher, # 25200–056) every 2 to 3 days. Human branching lung organoids were prepared using our normal human-induced pluripotent stem cell lines [52] with an established differentiation medium kit (STEMCELL Technologies, #100-0195/0528).

### Molecular biology constructs

Plasmid DNA constructs were generated using standard methods with restriction enzymes (New England BioLabs), DNA ligase (MightyMix, TaKaRa Bio/Clontech), and polymerase chain reaction (PCR) with Phusion polymerase (Thermo Fisher). Construct inserts for these experiments were synthesized (Integrated DNA Technologies, IDT) and subcloned into pcDNA3 vector (Life Technologies). Mock transfections were performed by using pcDNA3 empty vector.

### Imaging experiments and pH measurements

For the imaging experiments in Figs 1 and 7, NIH 3T3 cells were plated at a cell density of $2.5 \times 10^5$ cells/ml. The following day, the cells were transfected using Lipofectamine 3000 reagents (Thermo Fisher, # L3000001), and 4 μg of plasmid were mixed into 125 μl of serum-free OptiMEM with 5 μl of P3000 reagent. This was then added to another 125 μl of serum-free OptiMEM containing 7.5 μl of Lipofectamine 3000. Plasmid/P3000-lipofectamine complex was incubated for 15 min at room temperature, and then added to the plate. The medium was replaced 20 to 24 h after transfection, and 1 μM of Lysosensor Green DND-189 (Thermo Fisher, L7535) was added. The cells were incubated for 30 min in a 37˚C, 5% $CO_2$ incubator. The medium was replaced one final time prior to imaging. The live cell imaging was conducted on a customized/automated fluorescence microscope (Ti-E, Nikon) using an environmental chamber (TOKAI HIT) and the culture medium to maintain normal cell culture conditions (37˚C, 5% $CO_2$, 20% $O_2$). Transfection efficiency was estimated by counting cells that showed mKate2 red fluorescence and was typically between 25 and 35%. Fluorescence quantification and analysis was performed with ImageJ software and Prism 7/8/9 (GraphPad). Representative images were also gathered on a Leica DMi8 confocal microscope. The following peptides (>95% purity) were synthesized by Thermo Fisher/Pierce Custom Peptide team:

MY18: MYSFVSEETGTLIVNSVL
6-Arg (Arg)-MY18: RRRRRR-MYSFVSEETGTLIVNSVL
TAT-MY18: GRKKRRQRRRPPQ-MYSFVSEETGTLIVNSVL
Penetratin (Pen)-MY18: RQIKIWFQNRRMKWKK-MYSFVSEETGTLIVNSVL

## Global quantitative proteomics by mass spectrometry

For global quantitative proteomics of HEK 293S transfected using Lipofectamine 2000 (Invitrogen 11668027) with pcDNA3-SARS-CoV-2 Envelope (WT)-mKate2, tandem mass tag (TMT)-based quantitative proteomics was used. In brief, frozen cells were lysed by bead-beating in 9 M urea and 200 mM EPPS (pH 8.5), supplemented with protease and phosphatase inhibitors. Samples were reduced with 5 mM tris(2-carboxyethyl)phosphine and alkylated with 10 mM iodoacetamide that was quenched with 10 mM Dithiothreitol. A total of 100 μg of protein was chloroform−methanol precipitated. Protein was reconstituted in 200 mM EPPS (pH 8.5) and digested by Lys-C overnight and trypsin for 6 h, both at a 1:50 protease-to-peptide ratio. Digested peptides were quantified using a Nanodrop at 280 nm and 50 μg of peptide from each sample were labeled with 400 μg TMT reagent using 10-plex TMT kit [53]. TMT labels were checked, 0.5 μg of each sample was pooled, desalted and analyzed by short synchronous precursor selection (SPS) MS3 method, and using normalization factor, samples were bulk mixed at 1:1 across all channels and 500 μg of the bulk mixed sample was used for total proteome analysis.

Mixed TMT-labeled samples were vacuum centrifuged and desalted with C18 Sep-Pak (100 mg) solid-phase extraction column. The desalted sample was fractionated using BPRP chromatography. Peptides were subjected to a 50 min linear gradient from 5 to 42% acetonitrile in 10 mM ammonium bicarbonate (pH 8) at a flow rate of 0.6 ml/min over Water X-bridge C18 column (3.5 μm particles, 4.6 mm ID, and 250 mm in length). The peptide mixture was fractionated into a total of 96 fractions, which were consolidated into 28 fractions. Fractions were subsequently acidified with 1% formic acid, and vacuum centrifuged to near dryness and desalted via SDB-RP StageTip.

For total proteome analysis, 28 desalted fractions were dissolved in 10 μl of 3% acetonitrile/ 0.1% formic acid injected using SPS-MS3. The UltiMate 3000 UHPLC system (Thermo Fisher) and EASY-Spray PepMap RSLC C18 50 cm × 75 μm ID column (Thermo Fisher) coupled with Orbitrap Fusion (Thermo Fisher) were used to separate fractioned peptides with a 5% to 30% acetonitrile gradient in 0.1% formic acid over 45 min at a flow rate of 250 nl/min. After each gradient, the column was washed with 90% buffer B for 10 min and re-equilibrated with 98% buffer A (0.1% formic acid, 100% HPLC-grade water) for 40 min. The full MS spectra was acquired in the Orbitrap Fusion Tribrid Mass Spectrometer (Thermo Fisher) at a resolution of 120,000. The 10 most intense MS1 ions were selected for MS2 analysis. The isolation width was set at 0.7 Da and isolated precursors were fragmented by CID at normalized collision energy (NCE) of 35% and analyzed in the ion trap using "turbo" scan speed. Following the acquisition of each MS2 spectrum, a SPS-MS3 scan was collected on the top 10 most intense ions in the MS2 spectrum. SPS-MS3 precursors were fragmented by higher energy collision-induced dissociation at an NCE of 65% and analyzed using the Orbitrap.

Raw mass spectrometric data were analyzed using Proteome Discoverer 2.4 to perform database search and TMT reporter ions quantification. TMT tags on lysine residues and peptide N termini (+229.163 Da) and the carbamidomethylation of cysteine residues (+57.021 Da) was set as static modifications, while the oxidation of methionine residues (+15.995 Da), deamidation (+0.984) on asparagine and glutamine were set as a variable modification. Data were searched against a UniProt human with peptide-spectrum match and protein-level at 1% FDR. The signal-to-noise (S/N) measurements of each protein were normalized so that the sum of

the signal for all proteins in each channel was equivalent to account for equal protein loading. Results obtained from PD2.4 were further analyzed using Perseus statistical package [54] which is part of the MaxQuant distribution. Significantly changed protein abundance was determined by ANOVA with $P < 0.05$ (permutation-based FDR correction). Pathway analysis was performed using ingenuity IPA (Qiagen).

## Luciferase assay

HEK 293T cells were plated at $0.5 \times 10^5$ cells/well in 24-well plates (Corning) coated with poly-ornithine (Sigma-Aldrich). The following day, the cells were transfected with DNAs encoding the Envelope protein of interest, an NFAT *Firefly* Luc (NFAT-FLuc) reporter (using 4× NFAT site from human *IL-2* gene), and pRL-TK-*Renilla* Luc reporter (TK-RLuc, transfection control reporter using HSV *TK*, *herpes simplex virus thymidine kinase*, promoter) using Lipofectamine 2000 reagents. The standard transfection ratio of Envelope protein: NFAT-FLuc: TK-RLuc was as follows: 0.3 μg: 0.3 μg: 0.03 μg. Peptides of interest were added to this mixture at an amount of 0.1 μg. The cells were then incubated overnight at 37˚C in a $CO_2$ incubator. The following day, the cells were treated with 1 μM Phorbol 12-myristate 13-acetate (Sigma-Aldrich, P1585) for 8 h at 37˚C in a $CO_2$ incubator. Luc activity levels were then assayed using Dual Luciferase assay kit and Veritas 96-well luminometer (Promega, E1910) following the manufacturer's instructions. Following the luciferase results, TAT-MY18-2ED peptide (>95% purity) was synthesized by Thermo Fisher/Pierce Custom Peptide team:

TAT-MY18-2ED: GRKKRRQRRRPPQ-MYSFVS<u>DD</u>TGTLIVNSVL (M.W. = 3,662.2416).

## Western blotting

Western blot experiments were conducted using standard method. In brief, HEK 293T cells were plated in a 6-well dish at a $1 \times 10^6$ cells/well density. Cell samples receiving iPep-SARS2-E peptide (10 μm) were treated with the peptide for 24 h prior to transfection, and the peptides were refreshed during transfection. For transfection, 4 μg of the plasmid were mixed into 125 μl of serum-free OptiMEM, and 5 μl of Lipofectamine 2000 added to another 125 μl of serum-free OptiMEM. The 2 pots were combined and incubated at room temperature for 15 min. The next day, cells were lysed in cell lysis buffer (10×, Cell Signaling Technology, #9803) with 1% protease inhibitor cocktail (Sigma-Aldrich/Millipore-Sigma). Infected Vero-E6 cells were also treated and inactivated using the same lysis buffer. Samples were denatured using 2× SDS sample buffer (8 M urea, 40 mM Tris-Cl (pH 6.8), 2% SDS, 10% 2-mercaptoethanol) and boiled at 95˚C for 5 min. SDS-polyacrylamide gel electrophoresis was performed using home-made gels with Tris-Glycine (Bio-Rad) containing 10% Acrylamide-Bis (Fisher Scientific), 4% to 20% gradient or 16% pre-made gels (Novex, Thermo Fisher), which were then transferred to polyvinylidene difluoride (PVDF) membranes. Primary antibodies to anti-Spike S2 (R&D, MAB10557100, 1/5,000 dilution), anti-GFP (MBL#598, 1/8,000 dilution), and anti-GAPDH (rabbit recombinant monoclonal antibody, ab181602, 1:10,000 dilution, Abcam). Secondary antibody α-mouse (Thermo Fisher, #31430, 1/8,000 dilution) or α-rabbit (Thermo Fisher, #31460, 1/8,000 dilution) with 5% skim milk in Tris-buffered saline with 0.1% Tween 20 (TBS-T) was used for blocking of the PVDF membranes. Pierce enhanced chemiluminescence (ECL) western blotting substrate (Thermo Fisher, #32209) was used for the chemiluminescent reaction with films or ChemiDoc MP Imaging system (Bio-Rad).

## Monoclonal antibody production

Immunization of animals: The emulsion was prepared by mixing synthesized SARS2-E N-term peptide (Thermo Fisher) in 50% Dimethylsulfoxide (DMSO) in PBS and the adjuvant

complete freund (BD, #263810) evenly to make final peptide concentration of 1 mg/ml. Eight-week-old WKY/NCrl female rat purchased from Charles River Laboratories (Massachusetts, United States of America) was injected intramuscularly at the right and left tail base with 100 μl each of emulsion (total 200 μg peptide/rat) under our animal protocol (AC-AABC3508). Three times booster injections were done using the same method except at days 14, 17, and 20 where adjuvant incomplete Freund (BD, #963910) was used instead of complete Freund.

Lymphocyte harvest: As a partner cell, mouse myeloma cell line NS-1 (P3/NS1/1-Ag4.1, Sigma-Aldrich, # 85011427-1VL) was used. NS-1 cells were maintained with NS-1 medium (DMEM with 20% FBS, 1× GlutaMax supplement and 1× PS; Gibco #10313–021, HyClone #SH30071.03, Gibco #35050–061 and Gibco #15140–122, respectively) at 37°C, 5% $CO_2$. Two days after the last antigen injection, the rat was euthanized, and the medial iliac lymph node was taken aseptically and placed into 1 ml of NS-1 medium and cut to small pieces using sterile surgical blades. Following gentle pipetting to separate lymphocyte cells from other tissues, the lymphocyte cells were strained using a 100 μm pore cell strainer (Falcon, #352360). After counting, lymphocytes were frozen in NS-1 medium with 10% DMSO and kept in liquid nitrogen for storage.

Cell fusion: Lymphocytes were initiated the day before fusion and cultured in NS-1 medium. Fifteen million lymphocytes and 30 million NS-1 cells were mixed and spun down at 500xg for 2 min. After washing with HBSS (Gibco, #14175095), the cell pellet was resuspended in the fusion medium (0.3 M mannitol with 0.1 mM $CaCl_2$ and $MgCl_2$). The mixture was then put into the fusion chamber and fused using an electrofusion method (NAPA GENE, #ECFG21, for align; 30V for 20 s. For fusion; 350 V for 30 μsec 3 times with 0.5 s interval). The mixture was then collected from the fusion chamber and spun down at 500xg for 2 min. The fusion pellet was then resuspended to first culture medium (fresh NS-1 medium mixed with equal volume NS-1 cultured conditioned medium with 1× Hymax Hybridoma Fusion & Cloning Supplement (Antibody Research Corporation, Missouri, USA, #113004)) and plated on a 96-well plate. Hybridomas were selected by HAT sectioning culture for 2 weeks with NS-1 medium with 1× HAT media supplement (Sigma-Aldrich, #H0262) and 1× Hymax supplement, followed by HT maintenance culture with NS-1 medium with 1× HT media supplement (Sigma-Aldrich, #H0137) and 1× Hymax solution.

Screening: Primary screening was done by immunocytochemistry using 2E-YFP expressing plasmid transfected NIH 3T3 cells. The cells were plated on a Nunc Lab-Tek II chamber slide (Thermo Fisher, #154534) at a density of 15,000 cells/well and transfected with plasmid using Lipofectamine 3000 (Invitrogen, #L3000001) following the manufacturer's protocol. Twenty hours later, the cells were fixed using 4% paraformaldehyde in PBS. After 3 times wash with PBS, the cells were incubated with hybridoma culture supernatant with 0.5% NP-40 for 1 h at room temperature. After 3 times wash with PBS, the cells were incubated with anti-rat IgG secondary antibody conjugated with Alexa Fluor 594 (1:2,000 dilution, Abcam, #ab150160) for 30 min at room temperature. Positive clones were expanded to 24-well plate. After reaching 80% confluency, western blot was done as a secondary screening. For western blot preparation, SARS2-E-YFP plasmid was transfected into HEK 293T cells in a 10 cm dish using Lipofectamine 2000 following manufacturer's protocol. Twenty-four hours later, the cells were lysed using lysis buffer (Cell Signaling, #9803S) and spun down to collect the supernatant. The supernatant was mixed with same volume of the 2× SDS-urea sample buffer and boiled at 95°C for 5 min. Following the SDS page using 10% Bis-acrylamide gel, the protein was transferred to a PVDF membrane by electroblotting. Following 1 h blocking by 5% milk in TBS-T, the membrane was cut longitudinally into 0.5 cm wide strip. The strips were incubated with hybridoma culture supernatant for 1 h at room temperature. After 3 times washing with TBS-T, the strips were incubated with anti-rat IgG secondary antibody conjugated with HRP

(Invitrogen, #31470) for 30 min at room temperature. After 3 times washing, development was done using ECL solution following the manufacture's protocol. The anti-GFP antibody was used as a positive control. If positive, single-cell cloning was done using a limiting dilution method and another round of immunocytochemistry and western blotting was completed to confirm the result. Following immunocytochemistry and western blotting confirmation for the single cloned hybridomas, positive clones were passaged with gradually reduced concentration of Hymax supplement (1/2, 1/4, 1/10, and finally without Hymax) until the hybridomas could be maintained by NS-1 medium with 1× HT solution. Hybridoma isotyping was done using rat isotyping kit (Bio-Rad, #RMT1).

Antibody purification: 50 ml of hybridoma culture supernatant was collected followed by filtration using a 0.22 μm filter. The culture supernatant was mixed with an equal volume of 20 mM Na-phosphate solution (pH 7.0), with additions of NaCl (150 mM, final conc.) and Tween-20 (0.02%, final conc.) as well, and 0.25 ml of Protein-G agarose (Thermo Fisher, #20399) was added and gently rocked for 1 h at room temperature. Protein-G agarose was packed in a column using an open column method and washed with 20 ml of 20 mM Na-phosphate solution (pH 7.0); 100 mM Glycine-Cl solution (pH 2.7) was used as elution buffer and eluted solution was immediately neutralized by 1/10 volume of 1 M Tris-HCl solution (pH 8.0). The neutralized antibody solution was concentrated, and buffer changed to 20 mM Na-phosphate with 0.25 M NaCl solution using an Amicon Ultra-4 filter (size of 3K, Millipore, #UFC800396). Antibody concentration was measured using a rat IgG ELISA kit (Abcam, #ab189578).

### Electrophysiological recording

SARS2-E-PM-mKate2 construct was co-expressed with pcDNA3-MY18-2ED or pcDNA3 empty vector in HEK 293S cells as described previously [19]. Whole-cell patch-clamp recordings of the transfected mKate2-positive cells were conducted using a MultiClamp 700B patch-clamp amplifier (Molecular Devices) and an inverted microscope equipped with differential interface optics (Nikon, Ti-U). The glass pipettes were prepared using borosilicate glass (Sutter Instrument, BF150-110-10) using a micropipette puller (Sutter Instrument, Model P-97). Voltage-clamp measurements were conducted using an extracellular solution consisting of normal Tyrode solution containing 140 mM NaCl, 5.4 mM KCl, 1 mM $MgCl_2$, 10 mM glucose, 1.8 mM $CaCl_2$, and 10 mM HEPES (pH 7.4 with NaOH at 25°C) using the pipette solution: 120 mM K D-gluconate, 25 mM KCl, 4 mM MgATP, 2 mM NaGTP, 4 mM Na2-phospho-creatin, 10 mM EGTA, 1 mM $CaCl_2$, and 10 mM HEPES (pH 7.4 with KCl at 25°C). The recordings were conducted using the extracellular solution warmed at 37°C. The patch-clamp data was acquired and analyzed using pClamp 10 and Clampfit 10.4 (Molecular Devices).

### Immunoprecipitation

pcDNA3-SARS2-E-YFP with pcDNA3-6xHis-MY18-2ED or pcDNA3 empty vector were co-transfected to HEK 293T cells using Lipofectamine 2000. Twenty-four hours after the transfection, the cells were washed with PBS once and then treated with cell lysis buffer (Cell Signaling Technology). Standard purification method for His-tagged construct with Ni Sepharose 6FF (Sigma-Aldrich/Cytiva, #17-5318-01) was conducted using our established method as described previously [51].

### Peptide permeability assay

The peptides (>95% purity) were synthesized by Thermo Fisher/Pierce Custom Peptide team. Peptides were resolved in water and stored at a 2.5 mM stock concentration. For permeability

assay, NIH 3T3 and Vero-E6 cells were plated on 35 mm dishes at a density of $1 \times 10^5$ cells/ml. ON kinetics: peptides were added to cell culture medium at a concentration of 10 μM and incubated under normoxia conditions until time points for imaging. OFF kinetics: peptides were added to the cell medium at a concentration of 10 μM and incubated under normoxia conditions. After 24 h, the medium was replaced (no new peptide added). Cells were then imaged at determined time points. For all imaging, cell media was replaced with Tyrode's solution (140 mM NaCl, 5.4 mM KCl, 1 mM MgCl$_2$, 10 mM glucose, 1.8 mM CaCl$_2$, and 10 mM HEPES, pH buffered to 7.4 with NaOH).

Two types of Alexa Fluor 594-conjugated MY18-2ED peptides:
Alexa Fluor 594-TAT-MY18-2ED-:
Alexa Fluor 594-[C]G-GRKKRRQRRRPPQ-MYSFVS<u>DD</u>TGTLIVNSVL
TAT-MY18-2ED-Alexa Fluor 594:
GRKKRRQRRRPPQ-MYSFVS<u>DD</u>TGTLIVNSVL-L[C]-Alexa Fluor 594

## Stability assay of peptides

Peptide was synthesized by Thermo Scientific/Pierce and resolved in water at a 2.5 mM stock concentration and aliquoted for storage at −80˚C. Peptides were thawed on ice and diluted to 40 μg/ml using PBS. Samples were incubated at 37˚C. Baseline levels were measured using freshly thawed peptide diluted by 37˚C pre-warmed PBS. Samples were transferred to an EIA/RIA plate (Corning, #3591) at 50 μl/well and incubated overnight for coating at 4˚C. The next day, the plate was washed using 100 μl/well of PBS-T 3 times. The plate was blocked for 1 h at room temperature using 3% bovine serum albumin (BSA, Sigma-Aldrich) in PBS-T with 250 rpm shaking. Following this, the plate was washed 3 times using PBS-T. The primary antibody solution was then added at 50 μl/well (N2A5E8, 0.45 μg/ml) for 1 h at room temperature with 250 rpm shaking followed by 3 times wash with PBS-T. The anti-rat IgG secondary antibody HRP conjugated solution (1:10,000) was then added for 1 h at room temperature with 250 rpm shaking followed by 3 times wash with PBS-T. The development step was done using TMB solution (Thermo Fisher, #34021) following the manufacturer's manual. OD600 value was detected using a SpectraMax iD3 Plate Reader (Molecular Devices). Statistical analysis was done using GraphPad Prism software. This procedure was done in biological triplicate.

## Apoptosis/necrosis assay using peptides

Jurkat cells (ATCC, #TIB-152, clone E6-1) were cultured and treated for 48 h with 10 μM of iPep-SARS2-E peptide (TAT-MY18-2ED) or for 3 h with 10 μM (S)-(+)-camptothecin (positive control group, Sigma-Aldrich, #C9911). After treatment, the cells were collected for an Annexin-V assay (Thermo Fisher/Invitrogen, #V13241) following the manufacturer's manual. The data was collected by a ZE5 Cell Analyzer (Bio-Rad Laboratories) and analyzed using FlowJo software (BD Biosciences). This procedure was done in biological triplicate.

## Viral infection in vitro and sample processing

Cytopathic assay using WA10 virus (MOI, 0.10) in Vero-E6 cells was conducted as done in the previous study using standard methods [5]. Immunocytochemistry was conducted using a standard method using fixation solution containing 4% paraformaldehyde (Electron Microscopy Sciences) and 2% sucrose (Sigma-Aldrich) in PBS, blocking/permeability solution (2% BSA and 0.25% NP-40, Sigma-Aldrich, in PBS) and antibodies to ERGIC/p58 (Sigma-Aldrich, E1031), LAMP1 (Abcam, ab24170), BiP/GPR78 (Abcam, ab21685), and SARS-CoV-2 nucleocapsid (Thermo Fisher, PIMA17404). Goat anti-mouse IgG Alexa Fluor 594 antibody (Abcam, ab150116) and Goat anti-rabbit IgG Alexa Fluor 488 antibody (Abcam, ab150077) were used.

For electron microscopy, infected Vero-E6 cells were fixed using 2% paraformaldehyde, 2% glutaraldehyde (Electron Microscopy Sciences), and 2 mM $CaCl_2$ (Sigma-Aldrich) in 100 mM cacodylate buffer (pH 7.4, Electron Microscopy Sciences). For human branching lung organoids, EpCAM antibody (Abcam. ab223582) was used.

## Quantitative RT-PCR

RNA samples of Vero-E6 cells and mouse lung tissues were prepared using TRIzol Plus RNA Purification kit and PureLink DNase set (Thermo Fisher) while RNeasy Mini kit and RNase-Free DNase set (Qiagen) was used for HEK 293 cells. cDNA was synthesized using the Super-Script III First-Strand Synthesis System for RT–PCR (Thermo Fisher). *FAST* or *Power* SYBR Green PCR Master Mix and QuantStudio 3/7 real time PCR systems (Thermo Fisher) with StepOne software (version 2.3, Life Technologies) or CFX Opus 96 Real-Time PCR system (Bio-Rad) were used for qPCR using the below primer sets.

SARS-CoV-2 qPCR N forward primer: CTCTTGTAGATCTGTTCTCTAAACGAAC
SARS-CoV-2 qPCR N reverse primer: GGTCCACCAAACGTAATGCG
SARS-CoV-2 qPCR E forward primer: CTCATTCGTTTCGGAAGAGACAG
SARS-CoV-2 qPCR E reverse primer: AGACCAGAAGATCAGGAACTCTAG
Mouse Gapdh qPCR forward primer: CTTCACCACCATGGAGAAGG
Mouse Gapdh qPCR reverse primer: TGAAGTCGCAGGAGACAACC
Monkey qPCR GAPDH (for Vero-E6) forward primer:
GAAGGTGAAGGTCGGAGTCAAC
Monkey qPCR GAPDH (for Vero-E6) reverse primer:
TCGTTGTCATACCAGGAAATGAGC
Human NFATC4 qPCR forward primer: CTTCTCCGATGCCTCTGACG
Human NFATC4 qPCR reverse primer: CGGGGGCTTGGACCATACAG
Human JUN/AP-1 qPCR forward primer: ACTCGGACCTTCTCACGTC
Human JUN/AP-1 qPCR reverse primer: GGTCGGTGTAGTGGTGATGT
Human GAPDH qPCR forward primer: GATGACATCAAGAAGGTGGTGA
Human GAPDH qPCR reverse primer: GTCTACATGGCAACTGTGAGGA

## Viral infection in vivo and sample processing

Balb/c mice (8 to 11 weeks old, both male and female, Charles River) were infected intranasally with $5 \times 10^4$ PFU of SARS-CoV-2 (MA10) in a final volume of 50 μl (a single dose) with either mock (PBS), the negative control peptide, or iPep-SARS2-E treatment, following isoflurane sedation. After viral infection, mice were monitored daily for body weight, temperature, and foods. Mice showing >20% loss of their initial body weight were defined as reaching experimental endpoint and humanely euthanized before day 4. The peptides were provided intravenously (i.v., 2 mM, 150 μl in PBS, pH 7.0 adjusted with NaOH, a single dose), following a previous peptide-related study [9]. In case of a single dose of intranasal administration, the peptides, iPep-SARS2-E and the negative control mutant peptide, were used together with the infection under isoflurane sedation (2.5 mM, 50 μl in PBS, pH 7.0 adjusted with NaOH). The lung tissue samples were collected at the endpoint (4 days post-infection) for RNA preparation, lung histology, and/or lung viral titer using standard methods [9] as well as our optimized method of SARS2-E protein blotting described as the next section.

## Lung tissue western blotting

To inactivate the virus, MA10-infected mouse lung was incubated in 0.5% SDS in PBS for 1 h at room temperature. After being mashed by a plastic masher, the sample mixture was spun

down to collect the supernatant. The supernatant was mixed with same volume of 2× SDS-urea sample buffer and boiled at 95°C for 5 min. Following the SDS page using 20% Bis-acrylamide gel, the protein was transferred to a PVDF membrane by electroblotting. Following 1 h blocking by 5% skim milk in TBS-T, the membrane was incubated with the primary antibody solution (N2A5E8, 0.2 μg/ml in TBS-T) for overnight at 4°C. After 3 times washing with TBS-T, the membrane was incubated with anti-rat IgG secondary antibody conjugated with HRP for 1 h at room temperature. After 3 times washing, development was done using ECL solution following the manufacture's protocol.

## Fluorescent stereoscopic imaging of mouse tissues

iPep-SARS2-E peptide conjugated with Alexa594 at the C-terminus end (TAT-MY18-2ED-A594, 10 μM, 20 μl) or PBS was provided intranasally under the Isoflurane anesthesia. Two hours later, the mice were euthanized using $CO_2$. The skull was cut with sagittal section in the middle, followed by taking out the nasal septum. After 3 times wash in PBS, imaging for the lateral side of nasal cavity was conducted on a fluorescent stereoscope (Leica, M165 FC). Regarding i.v. injection method, iPep-SARS2-E group mice were injected with TAT-MY18-2ED-A594 peptide (300 μM, 50 μl) via tail vein before 24 h or 2 h of sacrifice. The control mouse was injected with PBS before 2 h of sacrifice. After sacrifice using $CO_2$ and cervical dislocation, whole body perfusion with 15 ml of PBS was conducted to wash out the peptides in their bloods. Harvested tissues, such as lungs, were briefly washed by PBS and imaging was conducted on the fluorescent stereoscope.

## Cytokine/Inflammation array

Cytokine inflammation panel was done in accordance with the manufacturer's protocol for the mouse cytokine array kit panel A (R&D Systems, Cat# ARY006). Serum samples used were collected from blood heat-inactivated at 65°C for 30 min (to inactivate any possible viruses) and spun down at 15,000 rpm for 10 min. Approximately 50 μl of blood serum samples were used for the assay. Cxcl12, C5a, MCSF, and CD54 were detected in the array using the denatured blood samples.

## Pseudo-virus design and production

Following a previous study [44], our pseudo viruses were designed and produced using HEK 293T cells transfected with pCMV-SARS-CoV-2 Spike (delta variant) and pcDNA3-M-P2A-E, pCMV-dR8.2 dvpr packaging plasmid (Addgene, #8455) and YFP reporter expressing plasmid, which was generated using standard PCR and LV-Cre-SD (#12105, no longer available at Addgene) with EcoRI and XhoI cloning sites to subclone YFP. The virus was filtered using a 0.45 μm syringe and then added to Vero-E6 cells (MOI, approximately 0.05). The YFP imaging was conducted 48 h post-infection using an epi-fluorescent microscopy (Nikon, TS100F).

## Statistics and reproducibility

The statistics used for every figure have been indicated in the corresponding figure legends. The Student's *t* test (paired and unpaired) was conducted with the *t* test functions in Microsoft Excel software. The Student's *t* test was two-tailed. The one-way ANOVA with Tukey's, Sidak's, Bonferroni's, or Dunnett's post hoc multiple comparison analysis was conducted with the GraphPad Prism 6/7/8/9 software. All the data meet the assumptions of the statistical tests. All the samples used in this study were biological repeats, not technical repeats. All experiments, except ELISA for the peptide-antibody binding assay (S2C Fig), were conducted using

at least 2 independent experimental materials/cohorts to reproduce similar results. No samples were excluded from the analysis in this study. All the graphs in the figures are mean ± SD.

## Supporting information

**S1 Fig. The effect of SARS2-E overexpression on mammalian transcripts.** Following the global proteomics results shown in Fig 2A, the expression of the gene transcripts was examined using qPCR. The expressions of *HSPA6* (**A**), *DAGLB* (**B**), *IP6K2* (**C**), *AGAP3* (**D**), and *RELB* transcripts (**E**) significantly increased in HEK 293S cells transfected to 2E-mKate2 compared to mock. The expression of the other genes, *TNC* (**F**), *CALU* (**G**), *PKLR* (**H**), *NOLC1* (**I**), and *ATF3* (**J**), did not significantly increase in the transfected HEK 293S cells though all the gene transcriptions slightly increased. Unpaired Student's *t* test was used (*** $P < 0.001$; ** $P < 0.01$; * $P < 0.05$; n.s., not significant, $n = 6$). The data underlying this figure can be found in S1 Data. All the graphs in the figure are mean ± SD.
(PDF)

**S2 Fig. Characterization of iPep-SARS2-E in situ and in vitro.** (**A**) Relative fluorescent intensity of DND-189 dye in NIH 3T3 cells transfected using mock ($n = 32$) or 2E-mKate2 plasmid without (-, $n = 27$) and with TAT-MY18-2ED peptide 24-h incubation (10 μM, $n = 50$). One-way ANOVA with Tukey's multiple comparisons test (**** $P < 0.0001$; n.s. not significant). (**B**) Representative immunoblot images of HEK 293T cells transfected with SARS2-E fused with YFP (2E-YFP). Anti-SARS2-E (2E-N, clone, N2A5E8, left), GFP (right), and GAPDH antibodies (as loading control, bottom) were used. The rat monoclonal antibody (2E-N mAb, clone N2A5E8) was produced using MY18 peptide conjugated to keyhole limpet haemocyanin (KLH) as the antigen. (**C**) ELISA assay for the comparison of binding capacity of the anti-SARS2-E monoclonal antibody to wild-type (WT) and 2ED (EE7-8DD) mutant TAT-MY18 peptides. (**D**) The stability test of iPep-SARS2-E (TAT-MY18-2ED) using the same ELISA assay with anti-SARS2-E antibody. The peptide was incubated at 37˚C in phosphate-buffered solution (PBS). (**E–G**) The toxicity test of iPep-SARS2-E (TAT-MY18-2ED, 10 μM, 48 h) using Jurkat cells and flowcytometry with apoptosis/necrosis assay. Healthy cells (**E**, %), apoptotic cells (**F**), and dead/necrotic cells (**G**) were counted. Camptothecin (10 μM, 3 h) was used as a positive control. One-way ANOVA with Dunnett's multiple comparisons test was used (**** $P < 0.0001$; n.s. not significant, compared to non-treated). (**H**) Immunoprecipitation of 2E protein using Ni column and HEK 293T cells transfected using 2E-YFP with 6xHis-MY18-2ED (2ED) or 6xHis-MY18 wild-type constructs (WT). Anti-GFP antibody was used to blot 2E-YFP protein bands. The data underlying this figure can be found in S1 Data. All the graphs in the figure, except S2C Fig, are mean ± SD. S2C Fig uses single-sample datasets.
(PDF)

**S3 Fig. The effect of iPep-SARS2-E on SARS2-E expression.** (**A**) Representative GAPDH immunoblot image of HEK 293T cells transfected with SARS2-E fused with YFP (2E-YFP) and treated with 10 μM TAT-MY18-2ED or MY18-WT (negative control). The image between 30 and 40 kDa is used in Fig 3E. (**B**) Representative immunoblot images of HEK 293T cells transfected with YFP plasmid and treated with 10 μM TAT-MY18-2ED or MY18-WT (negative control) for 48 h. Anti-GFP (for YFP, top) and GAPDH antibodies (as loading control, bottom) were used. The short- and long-exposure film images are shown. #, nonspecific bands around 40 and 50 kDa are found in the cell lysate. The band images are used in Fig 3H. (**C**) Quantification of YFP protein expression of HEK 293T cells transfected using YFP plasmid non-treated ($n = 3$) and treated with TAT-MY18-2ED ($n = 3$) or MY18-WT peptides ($n = 3$).

One-way ANOVA with Tukey's multiple comparisons test was used (n.s., not significant). The data underlying this figure can be found in S1 Data. The graph in the figure is mean ± SD. (PDF)

**S4 Fig. Permeability of iPep-SARS2-E.** (**A**) Representative fluorescent and bright field images of time-course cell-penetrating test using Alexa Fluor 594(A594)-conjugated iPep-SARS2-E peptides, A594-TAT-MY18-2ED (amino-terminal conjugation, N-term, 10 μM, bottom), and TAT-MY18-2ED-A594 (carboxyl-terminal, C-term, 10 μM, top) in NIH 3T3 cells after the incubation started. Scale bar, 50 μm. (**B**) Quantification of red fluorescence-positive cells treated with the A594-conjugated peptides for the peptide cell-penetrating "on" kinetics (mean ± SD). The data underlying this figure can be found in S1 Data. (**C**) Experimental design for the peptide stability, "off" kinetics, quantification. (**D**) Representative fluorescent and bright field images after washout of A594-conjugated TAT-MY18-2ED peptide (C-term version) in NIH 3T3 cells. White arrowheads, fluorescent puncta. Scale bar, 50 μm. (PDF)

**S5 Fig. Cell penetration of iPep-SARS2-E peptide in Vero-E6 cells.** (**A**) Representative red fluorescent and bright field images of Vero-E6 cells incubated with TAT-MY18-2ED-Alexa-Fluor594 (C-term conjugated version, 10 μM). Scale bars, 50 μm. (**B**) Quantification of red fluorescence-positive Vero-E6 cells treated with the AlexaFluor594-conjugated peptides (C-term conjugated version) for measuring the peptide cell-penetrating "on" kinetics (mean ± SD). The data underlying this figure can be found in S1 Data. (PDF)

**S6 Fig. iPep-SARS2-E in vitro validation.** (**A**) Electron microscopic image of iPep-SARS2-E-treated Vero-E6 cells at 24 h post-infection. Arrowheads, small particles found in the nuclear envelope. Scale bar, 1 μm. (**B**) Experimental design to examine whether intracellular particles are infectious in Vero-E6 cells treated with iPep-SARS2-E. (**C**) Result of quantitative endpoint titration assay used to quantify intracellular virus particles of PBS- and iPep-SARS2-E (10 μM)-treated Vero-E6 cells. There is a significant reduction of infectivity in iPep-SARS2-E though still infectious. Each well (cell plating at 24 h, 2,500 harvested cells onto $4 \times 10^4$ uninfected fresh cells per a well that were seeded the night before) was scored based on infectivity compared to virus controls with zero indicating no infection and 100 indicating complete infection (CPE). Student's $t$ test was used (**** $P < 0.0001$). (**D**) Western blots of Spike and GAPDH proteins in PBS- and iPep-SARS2-E (10 μM)-treated Vero-E6 cells at 24 h post-infection with SARS-CoV-2 WA1 (MOI, 0.10, 24 h), suggesting the effect of iPep-SARS2-E on Spike expression and/or stability. (**E–H**) qPCR of SARS-CoV-2 N (**E**), E (**F**), JUN/AP-1 expression (**G**) of PBS ($n = 6$)- and iPep-SARS2-E (10 μM, $n = 6$)-treated Vero-E6 cells comparing to non-infected cells ($n = 6$) at 48 h post-infection. The expression of these genes was normalized to a house-keeping gene, GAPDH. One-way ANOVA with Tukey's multiple comparisons test was used (**** $P < 0.0001$; *** $P < 0.001$; ** $P < 0.01$; n.s., not significant). (**H**) qPCR of SARS2-CoV-2 N expression of PBS ($n = 6$)- and iPep-SARS2-E (10 μM, $n = 6$)-treated Vero-E6 cell culture supernatant (sup) at 48 h post-infection. Student's $t$ test was used (**** $P < 0.0001$). The cDNA samples of cells and cell culture sup were prepared with TRIzol Plus RNA Purification kit, PureLink DNase set and SuperScript III and then diluted (1/5) using UltraPure distilled water for conducting qPCR. (**I**) Representative confocal fluorescent images of Vero-E6 cells treated with PBS or iPep-SARS2-E at 48 h post-infection. SARS-CoV-2 N antibody (red) and Hoechst 33258 dye (blue, for nucleus) were used with antibodies of subcellular organelle markers (green): BiP for endoplasmic reticulum (ER), ERGIC-53 for ER Golgi inter compartment (ERGIC), and LAMP1 for lysosome. Scale bar, 5 μm. The data underlying

this figure can be found in S1 Data. All the graphs in the figure are mean ± SD.
(PDF)

**S7 Fig. iPep-SARS2-E in vivo test using intranasal administration.** (**A**) Schematic representative of mouse intranasal administration of iPep-SARS2-E. Red box demonstrates the nasal tissue region harvested for the following fluorescent imaging. The image is from BioRender software. (**B**) Representative fluorescent and bright field images of nasal tissues isolated from mice administrated intranasally with PBS or Alexa594-conjugated iPep-SARS2-E peptide (TAT-MY18-2ED-A594, 10 μM, 2 h). After isolating the tissues, the samples were washing using PBS 3 times, and the fluorescent and bright field images were taken by a fluorescent stereoscope. Scale bar, 1 mm. (**C**) Experimental design for the iPep-SARS2-E safety test in vivo. (**D**) There is no significant difference the effects on body weight among iPep-SARS2-E-treated ($n = 6$), non-treated ($n = 5$), and PBS-treated Balb/c mouse groups ($n = 5$). One-way ANOVA with Tukey's multiple comparisons was used at each day. (**E, F**) There were no significant differences in Cxcl12 (**E**) and C5a (**F**) among iPep-SARS2-E-treated ($n = 5$), non-treated ($n = 4$), and PBS-treated mice ($n = 4$). One-way ANOVA with Tukey's multiple comparisons was used (n.s., not significant). (**G**) Experimental design for the iPep-SARS2-E test in vivo using intranasal administration. (**H**) iPep-SARS2-E prevents body weight loss in SARS-CoV-2 MA10-infected Balb/c mice ($5.0 \times 10^4$ PFU/mouse). Student's $t$ test was used at each day (** $P < 0.01$; * $P < 0.05$). (**I, J**) Representative immunoblots of SARS-CoV-2 E (2E, **I**) and mouse Gapdh proteins (**J**) in SARS-CoV-2 MA10-infected mouse lung tissues with PBS or iPep-SARS2-E treatment. (**K**) iPep-SARS2-E peptides significantly reduced the protein expression of 2E in MA10-infected Balb/c mouse lung tissues (PBS, $n = 4$; iPep-SARS2-E, $n = 4$). Student's $t$ test was used (* $P < 0.05$). The data underlying this figure can be found in S1 Data. All the graphs in the figure are mean ± SD.
(PDF)

**S8 Fig. iPep-SARS2-E and negative control peptide test in vitro and in vivo.** (**A**) Alignment of MY18 WT and mutant candidate sequences (MT, targeted amino acids, underline), following the result using the MY18 deletion constructs (Fig 2F) and mutagenesis. (**B**) Testing iPep-SARS2-E negative control (neg. Ctrl) mutant constructs using NFAT/AP-1 assay in mock- or 2E-transfected HEK 293T cells. One-way ANOVA with Tukey's multiple comparisons test (**** $P < 0.0001$; * $P < 0.05$; n.s., not significant). (**C**) Relative fluorescent intensity of DND-189 dye in NIH 3T3 cells transfected using mock or 2E-mKate2 plasmid without (-) and with the neg. Ctrl peptide constructs. One-way ANOVA with Tukey's multiple comparisons test (**** $P < 0.0001$; *** $P < 0.001$; n.s., not significant). (**D**) qPCR of SARS2 N expression of the neg. Ctrl mutant peptide-, iPep-SARS2-E-, and PBS- treated Vero E6 cell culture supernatant. All the peptides (10 μM) were used overnight (approximately 18 h) and then washed before SARS-CoV-2 WA1 infection. One-way ANOVA with Tukey's multiple comparisons test (** $P < 0.01$; n.s., not significant). (**E**) Representative images of phase contrast and yellow fluorescence of Vero-E6 cells infected with pseudo virus (MOI, 0.05) produced by SARS2 Spike, E, M, dR8.2 and YFP reporter using iPep-SARS2-E treatment (10 μM). The neg. Ctrl mutant peptide (10 μM) was used as a negative control. Scale bar, 20 μm. (**F**) There is no significant difference in YFP-positive cells between iPep-SARS2-E and neg. Ctrl, suggesting no effect of iPep-SARS2-E on the virus entry. Student's $t$ test was used (n.s., not significant). (**G**) Experimental design of the iPep-SARS2-E intranasal administration with the neg. Ctrl mutant peptide as a negative control in vivo. (**H**) Body weight changes of the mouse groups. Student's $t$ test was used (* $P < 0.05$). (**I**) There is a significant reduction of lung viral titer in iPep-SARS2-E-treated mice compared to the neg. Ctrl. Median tissue culture infection dose (TCID) is normalized to lung wet weight (g) measured before the tissue homogenization to isolate the virus.

(**J**) iPep-SARS2-E significantly reduced the transcript expression of SARS-CoV-2 N in MA10-infected Balb/c mouse lung tissues. Student's $t$ test was used (**** $P < 0.0001$; ** $P < 0.01$). The data underlying this figure can be found in S1 Data. All the graphs in the figure are mean ± SD.
(PDF)

**S9 Fig. Alignment of human coronavirus envelope proteins.** Envelope protein sequence alignment of SARS-CoV-2, MERS-CoV, HCoV-229E, HCoV-NL63, HCoV-OC43, and HCoV-HKU1. CLUSTALW 2.1 multiple sequence alignment software is used to obtain the alignment.
(PDF)

**S1 Data. Raw datasets of the experiments.**
(XLSX)

**S1 Raw Images. Western blotting original images.**
(PDF)

## Acknowledgments

We thank N. Harrison, R. Katz, M. Rahmany, C. Y.l. Sobolevsky, M.V. Yelshanskaya, C. Aston, E. Passague, and J. Stein (Columbia University) for their helpful support and discussion; Y. Tomono and K. Yamamoto (Shigei Medical Research Institute, Japan) for helpful advice with rat monoclonal antibody production; C. Castagna, Y. Luo, S. Sozomenu, A. Matveyenko and M.H. Blumenkrantz (Columbia University) and A. Poddar (Peddie High School, NJ) for helpful assistance.

## Author Contributions

**Conceptualization:** Masayuki Yazawa.

**Data curation:** Ramsey Bekdash, Kazushige Yoshida, Manoj S. Nair, Lauren Qiu, Johnathan Ahdout, Hsiang-Yi Tsai, Kunihiro Uryu, Rajesh K. Soni, Yaoxing Huang, Masayuki Yazawa.

**Formal analysis:** Ramsey Bekdash, Kazushige Yoshida, Lauren Qiu, Johnathan Ahdout, Hsiang-Yi Tsai, Yaoxing Huang, Masayuki Yazawa.

**Funding acquisition:** David D. Ho, Masayuki Yazawa.

**Investigation:** Ramsey Bekdash, Kazushige Yoshida, Manoj S. Nair, Lauren Qiu, Johnathan Ahdout, Hsiang-Yi Tsai, Kunihiro Uryu, Rajesh K. Soni, Yaoxing Huang, Masayuki Yazawa.

**Methodology:** Ramsey Bekdash, Kazushige Yoshida, Manoj S. Nair, Masayuki Yazawa.

**Project administration:** David D. Ho.

**Resources:** Manoj S. Nair, Yaoxing Huang, David D. Ho.

**Supervision:** David D. Ho, Masayuki Yazawa.

**Validation:** Ramsey Bekdash, Kazushige Yoshida, Masayuki Yazawa.

**Writing – original draft:** Masayuki Yazawa.

**Writing – review & editing:** Ramsey Bekdash, Kazushige Yoshida, Masayuki Yazawa.

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
