## [Editor Report · Decision Letter 0]

4 Oct 2023

Dear Dr. Yazawa, 

Thank you for submitting your manuscript entitled "Developing inhibitory peptide against SARS-CoV-2 Envelope" for consideration as a Research Article by PLOS Biology.

Your manuscript, the previous reviews and your responses to reviewers have now been evaluated by the PLOS Biology editorial staff, as well as by an academic editor with relevant expertise and I am writing to let you know that we would like to send your submission back to reviewers. We will try to send the manuscript back to the original reviewers, so they can assess your revision.

Once your full submission is complete, your paper will undergo a series of checks in preparation for peer review. After your manuscript has passed the checks it will be sent out for review. To provide the metadata for your submission, please Login to Editorial Manager (https://www.editorialmanager.com/pbiology) within two working days, i.e. by Oct 06 2023 11:59PM.

Kind regards,

Paula

---

Senior Editor

PLOS Biology

---

## [Decision Letter · Decision Letter 1]

6 Nov 2023

Dear Dr. Yazawa,

Thank you for your patience while your manuscript "Developing inhibitory peptide against SARS-CoV-2 Envelope" was peer-reviewed at PLOS Biology. It has now been evaluated by the PLOS Biology editors, an Academic Editor with relevant expertise, and by 2 of the original reviewers from the previous journal. 

In light of the reviews, which you will find at the end of this email, we would like to invite you to revise the work to thoroughly address the reviewers' reports.

As you will see below, the reviewers still have some concerns about your work and are not completely satisfied with the revised version. They think that further experimentation is needed to fully support the conclusions. Please address all the reviewers issues.

Given the extent of revision needed, we cannot make a decision about publication until we have seen the revised manuscript and your response to the reviewers' comments. Your revised manuscript is likely to be sent for further evaluation by all or a subset of the reviewers.

**IMPORTANT - SUBMITTING YOUR REVISION**

*Re-submission Checklist*

*Published Peer Review*

*PLOS Data Policy*

*Blot and Gel Data Policy*

Sincerely,

Paula

---

Senior Editor

PLOS Biology

REVIEWS:

Reviewer #1: The authors have developed a peptide based on SARS2 E protein that may have a therapeutic effect against COVID-19. This peptide (named iPepSARS2E) is based on the N-terminal 18 residues of E, and seems to inhibit lysosome acidification, whether through proton channel activity or membrane disruption, and also viral/host protein expression which is modified by E overexpression. The first is rationalized by a direct interaction of the peptide with E, and inhibition of its channel activity. The second, seems to be explained at least in part by increased degradation.

Negative controls in this new version are appropriate and some evidence is provided of physical interaction of E and the short peptide.

The authors show data both in vitro and in vivo that look intriguing. Overall, the paper shows that the N-terminal domain of the E protein in SARS-2 CoV reverses changes in lysosome fluorescence and host/viral protein expression caused by E overexpression or viral infection. I find the results showing the changes caused by the peptide convincing, although the mechanism by which these are happening are not entirely clear. It appears that the DD mutant has a direct interaction with E protein, but how is this linked to the events observed is more challenging to explain. The readouts used are sequence dependent: a double D mutant has a better effect than the WT, whereas another mutant chosen shows almost no effect. 

The DD mutant was found to have higher activity, and is used in subsequent experiments. If the hypothesis of interaction is correct, this mutant should have higher affinity for E than the WT sequence of the peptide, and even more for the negative control used. Can the authors show some experimental data that this is the case? By SPR/BLI for example?

The authors use electrophysiological recordings, but I am unable to comment if the current observed is due to E. The experiment may be far more convincing with a negative control E mutant which has no channel activity (N15A: refs PMID: 24788150, PMID: 22832120).

Minor issues, but important:

The authors use wrong citations. For example, line 82, in what way reference 11 is related to E oligomerization? The paper is a study of a small peptide at the C-terminal domain, and its interaction with another protein. 

In what way reference 14, which is just a review with a minor reference to E has contributed to E oligomerization knowledge? Please cite primary literature in which observations are first reported rather than reviews, in order to give credit where credit is due. But this requires to actually read the papers in question, not just the title.

The title of the paper is odd. "Developing inhibitory peptide against SARS-CoV-2 envelope". The authors are just careless in the use of language. The envelope of a virus is the lipid membrane that surrounds it, not a protein. They should refer in the title to the envelope PROTEIN. Grammatically, it is also wrong. Either 'Developing inhibitory peptideS' or 'Developing AN inhibitory peptide' would be correct. 

The two last paragraphs in the Discussion should be at least partially merged; they look repetitive (lines 356 and 363).

Reviewer #2: The authors addressed the reviewer's comments successfully except for the followings.

(Page 7 line 184) "…demonstrating a significant reduction of viral release from iPep-SARS2-E-treated cells." Based on the result in Fig. 5e, the authors cannot tell the decrease in N gene copy number is due to the reduction in viral release. This could be due to defect in virus assembly as well. If they want to distinguish virus assembly and secretion, they might need to quantify intracellular vs. extracellular infectious particles. In addition, measuring infectious virus titers is recommended rather than the qRT-PCR for quantification.

(Author response) We thank the reviewer for this advice. Following it, we have conducted further experiments to examine whether intracellular particles are infectious (Fig. S6b,c). The result suggests that the intracellular particles are still infectious. We really appreciate this reviewer's advice to improve our manuscript.

The authors addressed infectivity of intracellular particles by washing and trypsinization. However, it is questionable whether this procedure would release intracellular infectious particles successfully. Multiple freeze-thaw cycles would be a better method for releasing these intracellular particles.

(Page 11 line 288) "On the other hand, we found that there is no difference in SARS-CoV-2 nucleocapsid expression (Fig. 5d), suggesting no effect of iPep-SARS2-E on the viral entry." This statement needs more experimental evidences. For example, the authors need to conduct experiments with pseudovirus although in this case most of pseudovirus systems do not contain E protein.

(Author response) We appreciate this comment. Following it, we have conducted a new experiment using a pseudo virus containing E, M and S with YFP reporter to examine the effect of iPep-SARS2-E on the virus entry. We found no significant effect of iPep-SARS2-E on virus entry (Fig. R3/Fig. 8e,f). Following the new result, we have updated our discussion to explain the molecular mechanism underlying the effect of iPep-SARS2-E on SARS-CoV-2 function.

A pseudovirus containing SARS-CoV-2 S, M, E proteins was produced but no data was presented concerning the characterization of this pseudovirus. The evidence of presence and functionality of S, M, E proteins should be presented. Or the authors could cite any previous report in which this pseudovirus containing SARS-CoV-2 S, M, E proteins was described.

---

## [Editor Report · Decision Letter 2]

5 Jan 2024

Dear Masa,

Thank you for your patience while we considered your revised manuscript "Developing inhibitory peptides against SARS-CoV-2 Envelope protein" for publication as a Research Article at PLOS Biology. This revised version of your manuscript has been evaluated by the PLOS Biology editors and by the Academic Editor. 

Overall, we appreciates the additional efforts taken to address the reviewer concerns and while we understand that some of these efforts did not work, we are satisfied by the response to reviewers. Based on our Academic Editor's assessment of your revision, we are likely to accept this manuscript for publication. However, before we can accept your study, we need you to address the following data and other policy-related requests.

**IMPORTANT: Please address the following editorial requests. 

1) We think that some of your responses to Reviewer 1 in the rebuttal should be briefly included in the paper (or at least, we think it would be worthwhile to add some of the negative data provided in the rebuttal). 

2) TITLE: We note that "envelope" does not need to be capitalized in the title.

3) ABSTRACT: Please note that per journal policy, the model system/species studied should be clearly stated in the abstract of your manuscript. Please specify this more clearly. 

4) FINANCIAL DISCLOSURES: Please update the financial disclosures statement, in our online system, to include the grant numbers. Please also describe the role of any sponsors or funders in the study design, data collection and analysis, decision to publish, or preparation of the manuscript. If the funders had no role in any of the above, include this sentence at the end of your statement: "The funders had no role in study design, data collection and analysis, decision to publish, or preparation of the manuscript."

5) ETHICS STATEMENT: Please update the ethics statement, in your methods section, to specify which national/international regulations and guidelines your protocol adhered to. Please note that institutional or accreditation organization guidelines (such as AAALAC) do not meet this requirement.

6) BLURB: Please provide a blurb which (if accepted) will be included in our weekly and monthly Electronic Table of Contents, sent out to readers of PLOS Biology, and may be used to promote your article in social media. The blurb should be about 30-40 words long and is subject to editorial changes. It should, without exaggeration, entice people to read your manuscript. It should not be redundant with the title and should not contain acronyms or abbreviations.

7) DATA: Thank you for providing as a supplemental file, the data underlying your figures. Can you please reference this dataset in each relevant figure legend (including supplemental). For example, to each figure legend, you can add the sentence "the data underlying this figure can be found in ___"

8) WESTERN BLOTS: Thank you for providing, as a supplemental file, the uncropped blots related to the western blots presented in your paper. These generally look good to me - but they need to be annotated. Can you please update this file to annotate each blot with the following details: 

-which figure panel does the image relates to

- what sample was loaded in which well 

- what was the antibody used

- please indicate molecular weights of ladders

- please briefly note how image was collected

^Please be sure to make these changes without obscuring the western blot. 

^^For full guidelines for how to prepare and upload this data, see: https://journals.plos.org/plosbiology/s/figures#loc-blot-and-gel-reporting-requirements

We expect to receive your revised manuscript within two weeks. 

*Published Peer Review History*

*Press*

Sincerely,

Luke 

Lucas Smith, PhD

Senior Editor

PLOS Biology

lsmith@plos.org

---

## [Editor Report · Decision Letter 3]

25 Jan 2024

Dear Masa,

Thank you for the submission of your revised Research Article "Developing inhibitory peptides against SARS-CoV-2 envelope protein" for publication in PLOS Biology and thank you for addressing our editorial requests in this revision. On behalf of my colleagues and the Academic Editor, Frank Kirchhoff, I am pleased to say that we can in principle accept your manuscript for publication, provided you address any remaining formatting and reporting issues. These will be detailed in an email you should receive within 2-3 business days from our colleagues in the journal operations team; no action is required from you until then. Please note that we will not be able to formally accept your manuscript and schedule it for publication until you have completed any requested changes.

PRESS

Sincerely, 

Luke

Lucas Smith, Ph.D.

Senior Editor

PLOS Biology

lsmith@plos.org